# Sensitivity of simulating Typhoon Haiyan (2013) using WRF: the role of cumulus convection, surface flux parameterizations, spectral nudging, and initial and boundary conditions

Rafaela Jane Delfino[1,2], Gerry Bagtasa[2], Kevin Hodges[1], Pier Luigi Vidale[1]

[1] Department of Meteorology, University of Reading, Reading, United Kingdom
[2] Institute of Environmental Science and Meteorology, University of the Philippines-Diliman, Quezon City, Philippines

*Correspondence to*: Rafaela Jane Delfino (r.p.delfino@pgr.reading.ac.uk or rpdelfino@up.edu.ph)

**Abstract.** Typhoon (TY) Haiyan was one of the most intense and highly destructive tropical cyclones (TCs) to affect the Philippines. As such, it is regarded as a baseline for extreme TC hazards. Improving the simulation of such TCs will not only improve the forecasting of intense TCs but will also be essential in understanding the potential sensitivity of future intense TCs with climate change. In this study, we investigate the effects of model configuration in simulating TY Haiyan using the Weather Research Forecasting (WRF) Model. Sensitivity experiments were conducted by systematically altering the choice of cumulus schemes, surface flux options, and spectral nudging. In addition to using the European Centre for Medium-Range Weather Forecasts Re-analysis 5th Generation (ERA5) single high resolution realization as initial and boundary conditions, we also used four of the ten lower resolution ERA5 Data Assimilation System (EDA) ensemble members as initial and boundary conditions. Results indicate a high level of sensitivity to cumulus schemes, with a trade-off between using Kain-Fritsch and Tiedtke schemes that have not been mentioned in past studies of TCs in the Philippines. The Tiedtke scheme simulates the track better (with a lower mean Direct Positional Error (DPE) of 33 km), while the Kain-Fritsch scheme produces stronger intensities (by 15 hPa minimum sea level pressure). Spectral nudging also resulted in a reduction in the mean DPE by 20 km and varying the surface flux options resulted in the improvement of the simulated maximum sustained winds by up to 10 ms$^{-1}$. Simulations using the EDA members initial and boundary conditions revealed low sensitivity to the initial and boundary conditions, having less spread than the simulations using different parameterization schemes. We highlight the advantage of using an ensemble of cumulus parameterizations to take into account the uncertainty in the track and intensity of simulating intense tropical cyclones.

## 1 Introduction

As a country of 109 million people over more than 7,000 islands, the Philippines is considered one of the most natural hazard-prone countries in the world (Brucal et al., 2020) and is ranked in the top five of all countries in terms of exposure to climate-related risks (Eckstein et al., 2020). One of the most important hazards the Philippines is exposed to is tropical cyclones (TCs). TCs bring intense winds, extreme precipitation, and storm surges that affect a large portion of the Philippine population (Bagtasa, 2017; Lyon and Camargo, 2009). Due to its location in the western North Pacific Ocean, where TC formation is conducive all year, the Philippines is exposed to an average 10 landfalling TCs annually (Cinco et al., 2016). Since 1990, TCs in the Philippines have resulted in up to half of the total losses from all natural disasters amounting to about USD 20 billion in damages (Brucal et al., 2020) and an annual average death toll of 885 with estimated accumulated deaths due to TCs of approximately 30,000 from 1980 to 2013 (Yonson et al. 2016). It is estimated that about 5 million people are affected annually or over 570,000 are affected on average per destructive TC (Brucal et al., 2020).

One of the strongest typhoons that made landfall in the Philippines in recent history is Typhoon (TY) Haiyan (locally named "Yolanda"), which is considered the second costliest Philippine TC since 1990 (EM-DAT, 2020) and one of the deadliest since the 1970s (Cinco et al., 2016; Lander et al., 2014; Lagmay et al., 2015). TY Haiyan was a category-5 super typhoon that claimed the lives of at least 7,300 people, most of them from drowning due to the devastating 5 to 7-meter-high storm surge and coastal inundation (Soria et al., 2016). It also affected more than 16 million people (NDRRMC, 2014) and caused an estimated USD 5–15 billion worth of damage particularly in agriculture and critical infrastructure (Brucal et al., 2020). Comiso et al. (2015) found that TY Haiyan coincided with the warmest sea surface temperature (SST) observed over the Pacific warm pool region which may have contributed to its intense nature. This relation between intense TCs and warmer tropical SSTs, has also been found in the Atlantic (Emanuel 2005), and suggests that continuous warming may lead to more intense TCs in the future. Consistently, an increasing trend in intense TC frequency affecting the Philippines since the 1970's has been observed (Cinco et al., 2016; Comiso et al., 2015). TC rainfall is also expected to increase in the future as TCs intensify (Intergovernmental Panel on Climate Change, 2021; Patricola and Wehner 2018), potentially increasing the risk of flooding and landslides. Given TY Haiyan's intensity and impacts, it is regarded as a benchmark for an intense and destructive TC. Hence, it is important to test how well it can be simulated in current models in the present climate and the TC sensitivities to model formulation.

While global climate models (GCMs) are very useful for looking at the changes in TC activity under different climate change scenarios (e.g. frequency, intensity, genesis from a climatological and global/regional perspective) (Gallo et al., 2019; Patricola and Wehner 2018) and some advances have been made in the past few decades in the use of global convection-permitting models (Judt et al., 2021), previous studies still demonstrate the need for (convection-resolving/convection-permitting) limited area models (LAM) to better simulate the processes relevant to the TC formation and development as well as their properties, particularly the most intense ones (e.g. Walsh et al., 2013). In consideration of the computational cost in resolving important TC processes, the use of LAMs is a valuable and complementary approach to using GCMs in investigating the potential changes in TCs in the future. One such LAM is the Weather Research and Forecasting (WRF) Model (Skamarock et al. 2008), developed by the National Center for Atmospheric Research (NCAR), which is used as both numerical weather prediction LAM and regional climate model (RCM). WRF is currently used for operational forecasting in the Philippines by the country's meteorological office - Philippine Atmospheric, Geophysical and Astronomical Services Administration (PAGASA) (Flores 2019; Aragon and Pura 2016) and also used in hindcast simulation and sensitivity studies of TC track and intensity (Spencer and Shaw, 2012; Islam et al., 2014; Lee and Wu, 2018) and associated rainfall (Cruz and Narisma, 2016). It has also been used as an RCM to simulate TC activity in the WNP basin (Shen et al., 2017) and several TCs in the North Atlantic over a 13-year period in a convection-permitting model under current and future climate conditions (Gutmann et al., 2018). It has also been used as LAM in simulating specific TC cases with future GCM forcings as initial and lateral boundary conditions in other TC basins (Lackmann 2015; Parker et al., 2018; Patricola and Wehner, 2018).

WRF has also gained considerable popularity in recent years and has been used for TC simulations (Islam et al., 2015). Efforts are being made to identify the optimum parameterization schemes and to customize the WRF-ARW model for TC hindcast simulations. For instance, past NWP LAM studies of western North Pacific TCs, including TY Haiyan, show the cumulus (CU) convection scheme as having the most influence on its intensity over other model parameters such as the planetary boundary layer (PBL) and/or microphysics schemes (Islam et al. 2014; Di et al. 2019). In particular, the Kain-Fritsch (KF) (Kain, 2004) cumulus convection scheme has been found to produce the best TC tracks and wind intensity estimates (Zhang et al., 2011; Spencer and Shaw, 2012; Prater and Evans, 2002; Mohandas and Ashrit, 2012). Furthermore, the often-selected KF scheme was shown to be also sensitive to model resolution (Li et.al., 2018). However, the use of the KF scheme has also shown certain limitations. A study by Torn and Davis (2012) found that the KF scheme produces larger TC track biases than

the Tiedtke (TK) cumulus convection scheme.


Other than the said parameterization schemes, improvements in simulations of TC intensity have also been found to be influenced by the surface flux options (Kueh et al., 2019). Some other studies related to the sensitivity of WRF model choices can be found i.e. spectral nudging (Moon et al., 2018) and initial and boundary conditions (Islam et al., 2015). Previous work has explored the sensitivity of TC simulations in WRF to initial condition datasets i.e. from different reanalysis data (e.g.

Mohanty et al., 2010) and initial condition time (e.g. Mohanty et al., 2010; Shepherd and Walsh, 2016). Shepherd and Walsh (2016) showed that trajectories can be sensitive to initial condition time, however they are more sensitive to the CU parameterization. Mohanty et al. (2010) demonstrated that simulated intensity and vorticity maxima are sensitive to the chosen initial and boundary condition data set. Alternatively, nudging could be applied to the model until TC genesis, which would constrain the model to be more consistent with observations. Mori et al. (2014) applied spectral nudging in several runs in its

hindcast WRF simulations for Typhoon Haiyan and found that when applied, there is some bias in the simulated track primarily at landfall, but simulated reasonable intensities. Kueh et al. (2019) also performed several experiments with and without nudging at 3km resolution and found that nudging produced smaller track errors than the simulations without. They also found small differences in the TC intensity and structure in the experiments with and without nudging. Cha et al. (2011) suggested that continued spectral nudging can suppress TC intensification. Shen et al. (2017), although using WRF as an RCM in

investigating the effect of spectral nudging in inter-annual and seasonal variability of TC activity in East Asia, suggested that the nudging has an impact in reproducing TC activity. However, there are issues concerning the impact of nudging strength on model internal variability (Glisan et al., 2013). In this paper, we revisit the hindcast simulation of TY Haiyan using WRF as NWP LAM and assess its sensitivity to model formulation and the driving initial and boundary conditions, in preparation for pseudo-global warming and CMIP6 climate projection experiment studies. This study builds on the work of Islam et al.

(2015) who assessed the effects of different combinations of Planetary Boundary Layer, microphysics and cumulus convection scheme using WRF but found substantial underestimation of TY Haiyan's intensity regardless of the sensitivity to physics parameterization; Li et al. (2018) who used WRF to look at the effects of the cumulus parameterization at different resolutions (9-2 km) and found that the most effective resolution to simulate TY Haiyan with no cumulus parameterization or a revised KF scheme is at 2-km and 4-km resolution, respectively; and that of Kueh et al. (2019) who looked at the influence of the

different surface flux options in simulating TY Haiyan's intensity using one cumulus convection scheme and found that a better representation of surface flux formulas improved the simulated intensity in WRF. Here, we investigate the effects of the different combinations of model cumulus convection schemes, spectral nudging and surface flux options on the TY Haiyan track, intensity and rainfall hindcast simulations.

Improving the representation of intense TCs like Haiyan in LAMs such as WRF is also essential for simulations of such TCs in different future climate change scenarios to provide credible impact assessments and useful for simulating TC cases under different climate conditions e.g. pre-industrial or future (Parker et al., 2018; Patricola and Wehner, 2018; Chen et al., 2020). From this study the best combination is determined which will then be used for investigating the effects of future climate change on TY Haiyan and other TC cases. The associated storm surge of TY Haiyan (Mori et al. (2014), Nakamura et al.

(2017) and Takayabu et al. (2015) is not considered here. Model parameterization scheme sensitivity studies that assess the simulation of TCs will also provide guidance to future TC modelling studies (Villafuerte et al., 2021).

This study seeks to contribute to sensitivity studies with a particular focus on the Philippines by assessing the skill and sensitivity of a TC case study using a mesoscale NWP LAM model. In particular, it aims to study the influence of the

combination of cumulus convection scheme, the different surface flux options to the different TC characteristics and the use

of spectral nudging. This study adds on existing literature by looking at the effects of cumulus convection schemes combined with different flux options and spectral nudging. Specifically, it aims to address the following questions:

- How sensitive are the TY Haiyan hindcast simulations to convective schemes, surface flux options and spectral nudging?;
- How sensitive are the simulated track and intensity of TY Haiyan to the uncertainty in the initial and boundary conditions?

The results will provide valuable information for regional climate downscaling of intense TCs, which can be used in evaluating the sensitivity of future TCs in climate change simulations. Section 2 provides a description of the methodology. Then the paper continues with the results of the sensitivity experiments followed by the discussion, and finally, Section 4 provides a summary of the findings and recommendations for future work.

## 2 Method

### 2.1 Case Study: Typhoon Haiyan Brief Description

Typhoon Haiyan originated from an area of low pressure near the Federated States of Micronesia (157.2°E, 5.8°N) on November 2, 2013, and moved westward forming into a tropical storm on November 2, 2013. Typhoon Haiyan formed in an environment with a significantly high SSTs (peaking at 30.1°C in November 2013), which was considered the highest observed during the period between 1981 to 2014 in the Warm Pool Region (Comiso et al 2015). It then rapidly intensified into a TY on November 5 at 142.9°E, 6.9°N and was classified as a category-5 equivalent super typhoon by the Joint Tropical Warning Center (JTWC) and was classified as a Typhoon by PAGASA, its highest classification at the time. It further intensified before making landfall on November 7 2040 UTC. It traversed the central section of the Philippines and started to slowly weaken to a tropical depression on November 11 (JMA, 2013). Typhoon Haiyan claimed the lives of more than 7,300 people, mostly due to the associated storm surge and coastal inundation. It is estimated to have caused between USD 5-15 billion worth of direct damages in agriculture and infrastructure (Brucal et al., 2020) and affected more than 16 million people (NDRRMC 2014).

### 2.2 Model Description

Simulations were conducted using WRF version 3.8.1 (Skamarock et al. 2008), a non-hydrostatic numerical weather prediction LAM developed by the National Center for Atmospheric Research (NCAR). It is used for atmospheric research and operational forecasting, and increasingly for regional climate research (Powers et al., 2017). The model includes a variety of physical parameterization schemes, including cumulus convection, microphysics, radiative transfer, planetary boundary layer, and land surface. The WRF Advanced Research WRF (WRF-ARW) solver uses the Arakawa-C grid as the computational grid and the Runge-Kutta 3rd order time integration schemes (MMML-NCAR, 2019). Skamarock et al. (2008) provides a more detailed description of the model specifications. PAGASA uses WRF for its operational forecasting over the Philippine Area of Responsibility (Flores 2019; Aragon and Pura 2016) and it is also used in studies simulating event-based TC-associated rainfall over the Philippines (Cruz and Narisma, 2016).

The land surface information comes from the 30 arc s (*1 km) resolution Moderate Resolution Imaging Spectroradiometer (MODIS) satellite dataset with 20 global land use categories.

## 2.3 Initial and boundary conditions

The European Centre for Medium-Range Weather Forecasts (ECMWF) Re-analysis 5th Generation (ERA5) data is used for both the initial and boundary conditions. It is the latest generation of reanalysis products produced by ECMWF with horizontal resolution of 31 km, hourly temporal resolution and 137 vertical levels (Herbatsch et al., 2020). ERA5 uses observations collected from satellites and in-situ stations, which are quality controlled and assimilated using 4D-VAR, a model based on the ECMWF's Integrated Forecast System (IFS) Cycle 41r2.


Alongside the release of the ERA5 single realisation deterministic data from 1979 to the present, data from the Ensemble of Data Assimilations (EDA) system was also made available. The EDA system is a 10-member ensemble at a lower resolution than the deterministic data (60 km horizontal resolution and 3-hourly) (Hennermann, 2018). The EDA system provides estimates of analysis and short-range forecasts through one control and nine perturbed members, which provide background

error estimates for the deterministic forecasts. This system allows for estimating uncertainty since it provides estimates of the analysis and short-range forecast. These are provided as an uncertainty measure, albeit with half the resolution of the reanalysis. To test the sensitivity to varying boundary conditions, simulations were also performed using four randomly selected representatives of the 10-member ERA5 EDA system. The selected ensemble members were used to test the sensitivity to different perturbed observations, sea surface temperature fields and model physics (Isaksen et al., 2010).

## 2.4 Design of sensitivity experiments and analysis

In this study, the WRF–ARW model has been configured with two nested domains centered over the point of 18.3° latitude, 135° longitude. The outermost grid has 294 x 159 grid points with 25-km grid spacing, while the innermost domain has 745 x 550 grid points with 5-km grid spacing, with 44 vertical eta levels and the model top pressure level was set to 50 hPa. A two-way nesting is allowed for the interaction between the outer and inner domain. Specifically, for the outer domain which is

driven at the boundaries by ERA5, one-way nesting was used. For the inner domain, which is driven by the coarser domain, two-way nesting was used. The results shown in this paper are from the inner 5-km domain. This model resolution was chosen in favour of using supercomputing resources for the systematic testing of different parameterization schemes, and in consideration of additional simulations under future climate conditions.

Higher-resolution nested model configuration is widely used in numerical weather prediction and regional climate modelling. The main reason for this is because performing high resolution simulation over very large areas (e.g. an entire major oceanic basin) is computationally too expensive (Kueh et al 2019). The communication between the nested domains can be implemented using one-way or two-way nesting. One-way nesting means that the nested domains are run separately and sequentially starting with the outer domain i.e. the model is first run for the outer domain to create and output which is used to

supply the inner domain's boundary file. In a two-way nesting configuration, both domains are run simultaneously and interact with each other, so that the highest possible resolution information produced by the innermost domain affects the solutions over the overlapping area of the coarser domains. The input from the coarse outer domain is introduced through the boundary of the fine inner domain, while feedback to the coarse domain occurs all over the inner domain interior, as its values are replaced by combination of fine inner domain values (Alaka et al., 2022; Mure-Ravaud et al 2019; Harris and Duran 2009).

We have used two-way nesting in the sensitivity runs, rather than one-way nesting, following recommended practice and previous studies that looked at sensitivities to physics parameterizations in WRF (Wu et al., 2019, Biswas et al., 2014; Li and Pu, 2009; Parker et al., 2017; Spencer and Shaw 2012; Bopape et al., 2021), studies that simulated Typhoon Haiyan in the Philippines (Li et al., 2018; Nakamura et al., 2016), as well as TC cases in other basins (Parker et al., 2018; Mittal et al., 2019; Reddy et al 2020), among others. Studies of the differences in using 1-way and 2-way nesting in regional modelling have been,

the topic of multiple previous papers (e.g. Spencer and Shaw 2012; Matte et al. 2016; Raffa et al., 2021; Lauwaet et al. 2013;

Harris and Durran, 2010, Chen et al 2010; Gao et al., 2019). A comprehensive discussion on the differences and uncertainties associated with 1-way or 2-way nesting can also be found in Harris (2010). Studies such as those of Chen et al (2010) and Gao et al. (2019) have shown that the use of one-way or two-way nesting showed little difference in the results, but some studies showing that two-way nesting improves the simulations of TCs e.g. Typhoon Parma in the Philippines (Spencer and Shaw et al., 2012) and Typhoon Kai-tak (Wu et al., 2019). In addition, previous TC case studies in the Philippines have also used the two-way nesting configuration e.g. Mori et al. (2014), Takayabu et al. (2015), Nakamura et al (2016) and in other TC basins (Parker et al., 2018; Davis et al., 2008; Mittal et al., 2019; Reddy et al., 2020), as well as looking at sensitivity to different physics parameterizations (Wu et al 2019, Biswas et al 2014, Li and Pu 2009). Two-way nesting is also used in operational TC forecasting (Mehra et al., 2018) and in the experimental Hurricane WRF system (Zhang et al., 2016) as well as in Convection-Permitting Regional Climate Models (Lucas-Picher et al., 2021).

Different domain configurations were tested prior to selecting this particular configuration, with the current domain configuration having the track and intensity closest to that observed (Supplementary Figs. 1-5). The domain configuration used in this study is used to have common domain for different TC cases (other TC cases not included in this paper) to understand and have a more general set of conclusions on the response of TCs to future warming and to properly simulate the subtropical ridge/Western North Pacific Sub-tropical High.

In performing the experiments, WRF was run for a 180-hour period from 00 UTC 4 November 2013 – 12 UTC  11 November 2013 to cover the main part of the life cycle of TY Haiyan. Simulations with different start times were conducted (Supplementary Figs. 6-7) to sample the different stages in TY Haiyan's lifetime and different initializations. Starting times tested include 4 November 2013 at 00 and 12 UTC, 5 November 2013 at 00 and 12 UTC, 6 November 2013 at 00 and 12 UTC, and 7 November 2013 at 00 UTC. The simulation that started on 4 November 00 UTC was found to be optimal in terms of track and intensity, thus the initialization time of all experiments was fixed at 4 November 2013 00 UTC. The longer lead-time was also used to allow for the simulation of the early stages of development of Typhoon Haiyan. We considered the period covering 4 November 2013 00 UTC to 5 November 2013 12 UTC as the spin-up period. For the purposes of this paper, the analysis of the experiments covered only the 72-hour period between 18 UTC 5 November 2013 to 18 UTC 8 November 2013 to cover TY Haiyan's mature stage.

Additional simulations using convection-permitting resolution (single domain, 4.5km) were also performed and showed no significant change in simulated intensity from the configuration used here (not shown). The results shown in this paper are from the inner 5-km domain, with results of the outer 25-km domain shown in Supplementary Figure 8. The model domain set-up is shown in Fig. 1.

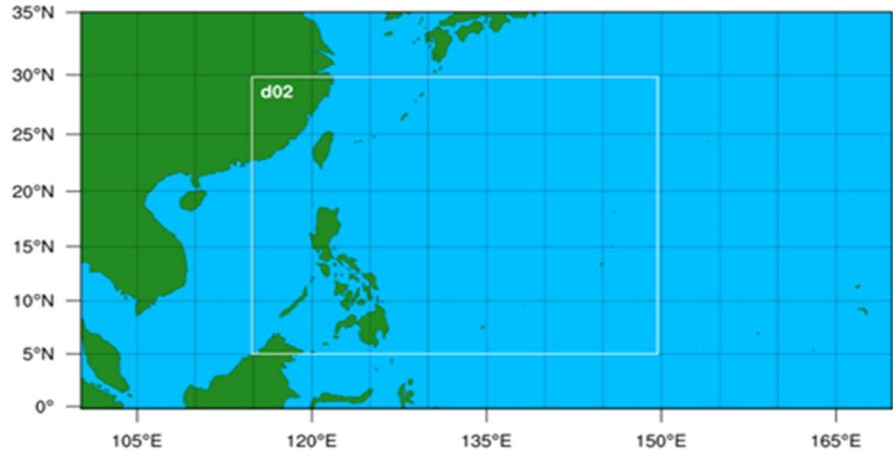


**Figure 1: Study domain set-up. The outer 25-km resolution (△x) domain is bounded by 100-170 degrees East and 0-35 degrees North while the inner 5-km resolution (△x) domain is bounded by 115-150 degrees East and 5-30 degrees North.**

Convection is mostly simulated in models with resolution coarser than 10-5km through the cumulus parameterization scheme.

WRF's cumulus parameterization scheme simulates the effects of cumulus convection on heat, moisture, and precipitation at the sub-grid scale (Skamarock et al. 2008). The choice of cumulus parameterization schemes has an impact on WRF's ability to simulate the TC track, intensity, and structure (Zhang et al., 2011; Shepherd and Walsh 2016; Parker et al., 2017). Only two schemes were investigated in this study the KF scheme and TK scheme, the difference of which are summarized below in Table 1. The same physics paramaterizations, including the cumulus scheme, were used in both inner and outer domains.

PAGASA uses KF for its operational forecasting configurations (Flores 2019). It has also often been used for TC simulation studies in the Philippines and has been found, in several studies, to be the best choice for simulating TC track and intensity (e.g. Sun et al., 2015; Li et al., 2018) and rainfall (e.g. Cruz and Narisma., 2016). The TK scheme, on the other hand, has been suggested to be the more appropriate cumulus scheme in tropical weather/climate applications of the WRF model (Parker et al., 2017). Torn and Davis (2012) showed an improvement in TC track simulations when using the TK scheme compared to

the KF scheme. They stated that the TK scheme allows for more appropriate treatment of oceanic shallow convection due to a more active shallow convection scheme than that of the KF scheme. There was a 1K temperature bias at 700 hPa in the KF scheme not present in the TK simulations, attributed to a lack of shallow convection in KF. These generated horizontal temperature gradients are associated with the wind biases affecting the TC tracks simulated with the KF scheme (Parker et al 2018; Torn and Davis, 2012). In addition, according to Sun et al. (2015), deep convection in mass flux schemes, such as KF,

produces large amounts of anvil clouds that warm the upper troposphere and cause latent heating south of the WNPSH that leads to the weakening of the WNPSH and the movement of the TCs northward. Li et al. (2018) investigated the sensitivities of the simulated tracks, intensities and structures of Typhoon Haiyan to the use of a the revised KF scheme with varying from resolutions from 9 to 2 km and found that the resulting simulations with the application of rKF scheme is different on various resolutions. Cruz and Narisma (2016) also used the KF scheme in conducting sensitivity tests of TC-associated rainfall with

different PBL and microphysics schemes in WRF.

Using a mass flux approach with downdraft removal, and utilizing Convective Available Potential Energy (CAPE), KF is a deep and shallow convection sub-grid scheme that includes clouds, rain, ice, and snow detrainment and cloud persistence (Kain 2004). Although KF can account for relatively small-scale processes that drive convection, it has inherent limitations in

simulating shallow convection over tropical oceans (Parker et al., 2017). On the other hand, the TK scheme assumes that the moisture flux through the cloud base is equivalent to the surface moisture flux, as well as momentum transport, cloud detrainment, and ice detrainment (Tiedtke 1989; Zhang et al., 2011). According to Parker et al. (2017), the TK scheme is more appropriate for simulating intense TCs in tropical oceans.

Table 1: Description of the cumulus schemes used in this study

| | **Kain-Fritsch (KF)** (Kain, 2004) | **Tiedtke (TK)** (Tiedtke 1989, Zhang et al., 2011) |
|---|---|---|
| Type of scheme | Mass-flux | Mass-flux |
| Cloud detrainment | Yes | Yes |
| Closure | CAPE Removal | CAPE / Moisture convergence |
| Triggering mechanism | Controlled by large scale velocity in the vertical direction | Convection is triggered if the parcel is warmer than its surroundings by 0.5K if the parcel is very close to the surface |
| Cloud radius | Variable | Fixed |
| Shallow convection | Activates shallow convection when the criteria for deep convection are satisfied | Assumes that the cloud base moisture flux is equal to the surface moisture flux |

Sources: Adenyi et al., (2019), Torn and Davis (2012), Shepherd and Walsh (2016)

Experiments were also conducted to examine the sensitivity to the available parameterizations for surface flux options. For TC applications, WRF-ARW provides three different formulations of aerodynamic roughness lengths of the surface momentum and scalar fields as surface flux options (isftcflx = 0,1, and 2) (see Kueh et al. 2019 for a detailed description of the differences between these options). It has been shown that surface fluxes can influence the model's ability to simulate TC intensity and structure (Green and Zhang, 2013; Kueh et al., 2019). For the default flux option (referred to here as sf0), the the momentum roughness length is given as Charnock's (1955) expression plus a viscous term, following Smith (1988) - Eq. (1):

$$z_O = \alpha \left( \frac{u^2}{g} \right) + \frac{0.11v}{u} \ , \tag{1}$$

where $\alpha$ is the Charnock coefficient and v the kinematic viscosity of dry air, for which a constant value of $1.5 \times 10^{-5}$ m2 s−1 is used. A constant value of α=0.0185 is used for sf0.

Since the roughness length formulas in sf0 are demonstrably inconsistent with a substantial amount of research (Kueh et al. 2019), two more options were developed (hereinafter referred to as sf1 and sf2) (Kueh et al. 2019). Based on the findings that the drag coefficient (CD) seemed to level off at hurricane force wind speed (e.g., Powell et al., 2003; Donelan et al., 2004), the surface flux option 1 (sf1) was developed and implemented in WRF as a blend of two roughness length formulas (Green and Zhang 2013). The sf1 option was first implemented in version 3.0 of WRF (Kueh et al. 2019). The sf1 and surface flux option 2 (sf2) has the same momentum roughness length, but in sf2, the temperatures and moisture roughness lengths are expressed in accordance with Brutsaert (1975) (MMML-NCAR, 2019). There are limited studies on the sensitivity of TC intensity due to surface heat flux because to a lack of in-situ measurements (Montgomery et al., 2010; Green and Zhang, 2013; Smith et al., 2014), particularly under high-wind conditions (Liu et al., 2014). Emanuel (1986) put forward the idea that TC intensity is proportional to the square root of the ratio of the surface exchange coefficients of enthalpy, and momentum. According to Zhang et al. (2015), increasing surface friction would also increase boundary layer inflow, which would subsequently boost angular momentum convergence and intensify a TC. However, as surface friction also increases the momentum and heat dissipation to boundary layer winds, this might result in negative impact on TC intensity (Liu et al., 2014). Despite and playing a significant role in surface heat fluxes, Chen et al. (2018) hypothesized that the influence of on TC growth was minimal because it caused moderate sea-surface cooling. Further investigation on these aspects is required in the future.

A set of experiments is conducted to explore the impacts of nudging on the ERA5 large-scale environment by applying spectral nudging (snON). It has been shown that spectral nudging can improve TC track simulations (Guo 2017; Tang et al., 2017) by constraining the model to large scale environmental conditions (Gilsan et al., 2013). Present-day simulations typically use nudging to reduce the mean biases in a relatively large domain (e.g., Xu and Yang 2015; Liu et al., 2016; Shen et al., 2017; Moon et al., 2018). Another set of experiments were also conducted without applying this technique (snOFF). Based on the methodology of Moon et al. (2018), the spectral nudging for the horizontal and vertical wind components, the potential temperature, and the geopotential height was applied. The nudging coefficients for all variables were set at 0.0003 s$^{-1}$, applied at all levels above the PBL.

To assess the model sensitivity to various physics parameterizations and other model choices, we have systematically altered the choice of cumulus schemes and surface flux options. The use of spectral nudging is also explored in a set of experiments. Table 2 shows the set of different model configurations.

**Table 2: Summary of the sensitivity experiments with the parameterizations used**

| Cumulus scheme | Nudging | Surface flux option (isftcflx)* | | |
|---|---|---|---|---|
| | | *isftcflx* = 0 (sf0) | *isftcflx* = 1 (sf1) | *isftcflx* = 2 (sf2) |
| KF (KF) | Without spectral nudging (snOFF) | KFsnOFFsf0 | KFsnOFFsf1 | KFsnOFFsf2 |
| | With spectral nudging (snON) | KFsnONsf0 | KFsnONsf1 | KFsnONsf2 |
| TK (TK) | Without spectral nudging (snOFF) | TKsnOFFsf0 | TKsnOFFsf1 | TKsnOFFsf2 |
| | With spectral nudging (snON) | TKsnONsf0 | TKsnONsf1 | TKsnONsf2 |

The control simulation is the experiment with KF as cumulus scheme, with spectral nudging turned off and surface flux option of sf0 (KFsnOFFsf0). This configuration was also used in the experiments using the different members of EDA, to test the sensitivity to different initializations.

Other parameterization schemes (adapted from Li et al., 2018) in the model that remained the same in all the experiments, as used in both inner and outer domains, include: the Rapid Radiative Transfer Model (RRTM) scheme (Mlawer et al., 1997) and the Dudhia scheme (Dudhia 1989) for the longwave and shortwave radiation, respectively; surface layer is from the MM5 Monin- Obukov scheme (Monin and Obukhov 1954); WRF Single–moment 6–class Scheme for the cloud microphysics (Hong and Lim 2006); and Yonsei University (YSU) PBL scheme (Hong et al., 2006); and the Unified Noah Land Surface Model (Chen and Dudhia 2001; Tewari et al. 2004) for the land surface processes and structure, as indicated in Table 3.

**Table 3 WRF Configuration for the control experiment (KFsnOFFsf0)**

| | |
|---|---|
| Number of Domain | Two (outer coarse domain D01 & inner domain D02) |
| Nesting | Two-way (between D01 & D02) |
| Grid resolutions | 25 km (D01); 5 km (D02) |
| Grid spacing | 295 x 160 (D01), 746 x 551 (D02) |
| Number of vertical eta levels | 44 (D01), 44 (D02) |

| | |
|---|---|
| Cloud microphysics | WRF Single–moment 6–class Scheme for the cloud microphysics (Hong and Lim 2006) – D01&D02 |
| Cumulus parameterization | Kain–Fritsch scheme – D01&D02 |
| Longwave radiation | RRTM scheme (Rapid Radiative Transfer Model) (Mlawer et al., 1997) |
| Shortwave radiation | Dudhia scheme (Dudhia 1989) – D01&D02 |
| Surface layer | MM5 Monin- Obukhov scheme (Monin and Obukhov 1954) – D01&D02 |
| Land surface scheme | Unified Noah Land Surface Model (Chen and Dudhia 2001; Tewari et al. 2004) – D01&D02 |
| Planetary boundary layer scheme | Yonsei University (YSU) PBL scheme (Hong et al., 2006) – D01&D02 |
| Surface flux option | isftcflx = 0 |
| Spectral nudging | Off |

## 2.5 Verification Data

To determine the model's skill in simulating TY Haiyan, we used the International Best Track Archive for Climate Stewardship (IBTrACS) which compiles best-track information from various agencies worldwide (Knapp et al., 2010). We compared the simulated and observed tracks by calculating the Direct Positional Error (DPE). Heming (2017) defines DPE as a measure of the great circle distance between observed and forecast positions at the same simulation time. We calculated the model bias, root-mean-square error (RMSE), and correlation coefficient between model-simulated and observed (IBTrACS) minimum sea level pressure, maximum 10m winds to evaluate simulated TC intensity. The best-track information used here is taken from the World Meteorological Organization (WMO) subset of the IBTrACS (IBTrACS-WMO, v03r09) which was taken from the best-track data provided by the Japan Meteorological Agency (JMA). In order to directly compare the IBTrACS / JMA data with WRF's simulated winds, the 10-min averaged winds from the JMA dataset were converted to 1-min wind speeds using Li et al.'s (2018) formula i.e. multiplying the 10-min values by 1.1364.

In addition, rainfall data from the Global Precipitation Measurement (GPM) is also used for comparing the spatial distribution of the simulated rainfall. The Integrated Multi-satellitE Retrievals for GPM (IMERG) is a third-level precipitation product of GPM, which covers the area -180, -90, 180, 90 with resolutions of 0.1 degree and 30 minutes (Huffman et al. (2019). The rainfall data were accessed and downloaded from NASA's Goddard Earth Sciences Data and Information Services Center (GES DISC) at https://disc.gsfc.nasa.gov/datasets/.

## 2.6 TC Tracking Method

The simulated track and intensity values were obtained every 6 hours using the TRACK algorithm (Hodges et al., 2017) as used in Hodges and Klingaman (2019). TRACK determines TCs as follows: first the vertical average of the relative vorticity at 850-, 700-, and 600-hPa levels is obtained. The field is then spatially filtered using 2D discrete cosine transforms equivalent to T63 spectral resolution and the large-scale background is removed. The tracking is performed by first identifying the relative vorticity maxima > $5.0 \times 10^{-6}$ s$^{-1}$. Using a nearest neighbour method, the tracks are then initialized and refined by minimizing a cost function for track smoothness subject to adaptive constraints (Villafuerte et al., 2021). The feature points are determined by first finding the grid point maxima which are then used as starting points for a B-spline interpolation and steepest ascent maximization method, to determine as the off-grid feature points (Hodges 1995 as cited by Hodges and Klingaman, 2019).

The tracking is done for the entire simulation period. Additional variables are added to the track data after the tracking is complete, such as the maximum 10-m winds within a $6^0$ geodesic radius and the minimum sea level pressure (MSLP) within a $5^0$ radius using the B-splines and minimization method (Hodges and Klingaman, 2019).

## 3. Results and Discussion

### 3.1 Simulated track

Fig. 2 shows the tracks obtained from the simulations of TY Haiyan for all experiments are in reasonably good correspondence with the best-track data. Simulations using the TK scheme accurately reproduced the observed positions of TY Haiyan during the first 36 h of the study period, with the observed and simulated tracks being less than 50 km (mean of 18km) apart at 36 h. On the other hand, the simulated tracks based on the KF cumulus convective scheme tracked in the same direction as the observed track, but were further north and more than 50 km (mean of 61.5km) from the best track during the first 36 hours of

simulation.

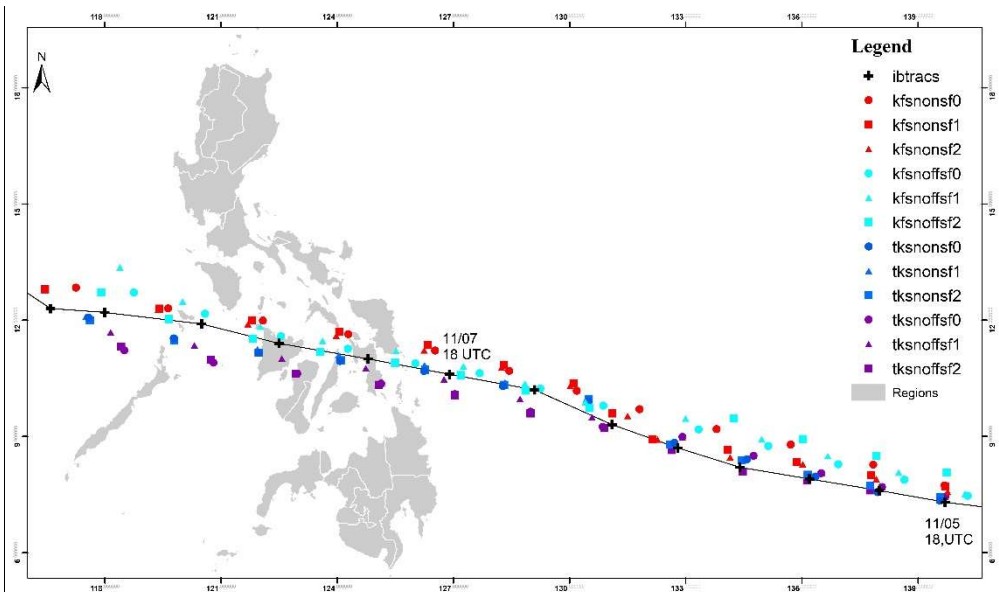

**Figure 2: Simulated tracks compared with IBTRaCS and the sensitivity experiments classified according to experiment groups:**
**Kain-Fristch (KF) convection scheme; TK (TK) convection scheme; with spectral nudging (snON); without nudging (snOFF); surface flux option 0 (sf0); option 1(sf1); and option 2 (sf2).**

Fig. 3 shows the sensitivity of the tracks to the cumulus parameterization scheme, surface flux options, and to spectral nudging. Fig. 3a shows the DPE throughout the simulation and shows simulations with the KF scheme have tracks that are further north

of the observed track compared to simulations utilizing the TK scheme which are closer to the observed track. The minimum DPE obtained from the simulations using the TK scheme is 8km after 18 hours of simulation for the simulation using TKsnOFFsf2.

The results show that these three model settings individually lead to significant reductions in DPE values. The difference

between the mean DPE of simulations using KF and TK schemes (p-value: 0.010) were found to be statistically significant at 99% confidence levels using a Student's t test. The simulations using the TK scheme have a mean DPE of $47 \pm 5$ km and KF scheme have mean DPE of $55 \pm 7$ km (Fig. 3a). Overall, we found the TK scheme to be best in simulating the track of TY Haiyan.

Our results show that the tracks are also slightly sensitive to the use of spectral nudging, especially in the latter half of the simulation (Fig. 3b). The evolution of DPE in Fig. 3 shows gradual increases in its value in the first half of the simulation, as the typhoon approaches land (between 48 h-54 h), the DPE then starts to abruptly increase until the end of the simulations. This suggests that the spectral nudging configuration does not constrain the model strongly. Nevertheless, simulations run with spectral nudging consistently show lower DPE in the 2nd half of the simulation compared to the no nudging experiments.

Moreover, the mean DPE of the TK simulations with nudging is 38 km while the simulation without nudging is 57 km. This is consistent with previous studies where spectral nudging improves TC tracks in the WNP (Guo 2017; Moon et al., 2018). Overall, the surface flux options did not have a statistically significant effect (p-value: 0.8509 at 95% confidence level) on the tracks of the simulated TY (Fig. 3c).


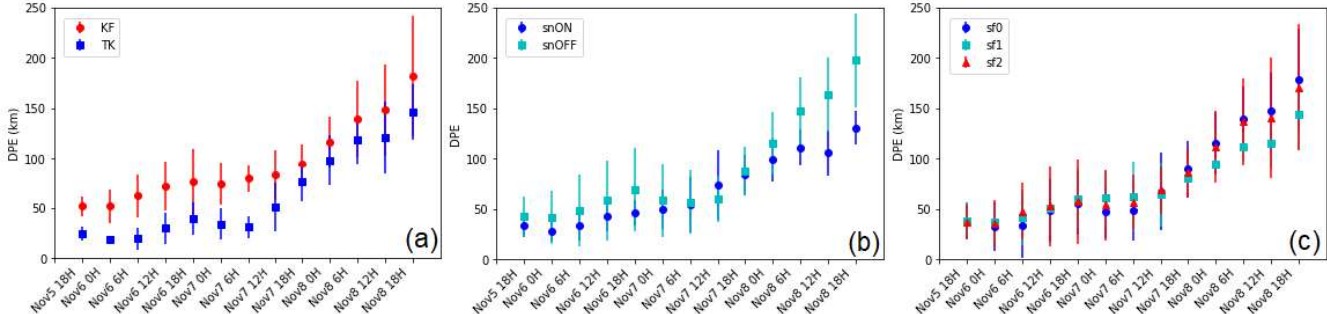

**Figure 3: Mean and standard deviation of the DPE (km) per simulation group – (a) for the cumulus schemes KF and TK, (b) for with (snON) and without nudging (snOFF) and (c) for surface flux options sf0, sf1 and sf2. X-axis is the analysis period between 18 UTC 5 November 2013 to 18 UTC 8 November 2013.**


## 3.2 Simulated intensity

Fig. 4 shows that most of the simulations are not able to capture the observed deepening of the minimum central pressure or the intensification of low-level winds of TY Haiyan. The control simulation (denoted as KFsnOFFsf0) has a MSLP value of only 939 hPa and maximum wind speed of 48.21 meters per second ($ms^{-1}$). Compared to the minimum central pressure of

895hPa in the observations, this is a difference of 44 hPa; and with the and 73 $ms^{-1}$1-min sustained wind speed, there is a difference of 24.79$ms^{-1}$ with the observed. The simulations that are closest to TY Haiyan's intensity are those that use the KF scheme and surface flux option 1 (KFsnONsf1), however, the simulations using KF scheme simulate lower than observed MSLP value at the first 12 hours of simulation. The KFsnONsf1 run has a MSLP reaching to 912 hPa and winds of up to 72 $ms^{-1}$. The TK scheme simulations consistently have higher central pressure and lower maximum wind speeds. A Student's t

test indicates that difference between the minimum sea level pressure simulations using KF and TK schemes (p-value: 0.008) is significant at the 99% confidence level. However, the simulations were not able to capture TY Haiyan's rapid intensification phase as in previous studies (Islam et al., 2015; Kueh et al., 2018).

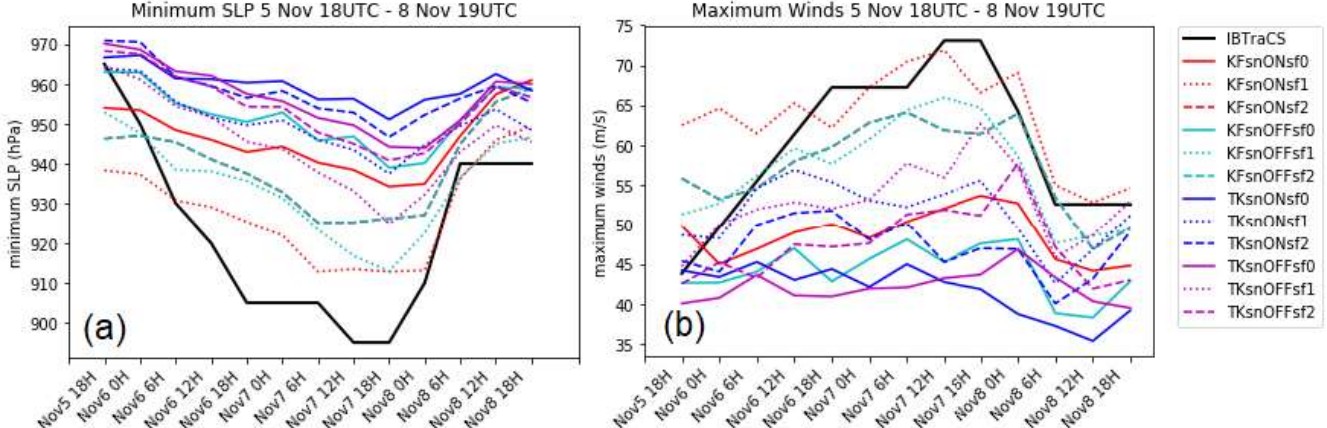


**Figure 4: Time series of intensity (a) for minimum sea level pressure in hPa and (b) maximum winds (m/s) for the sensitivity experiments classified according to experiment groups: Kain-Fristch (KF) convection scheme; TK (TK) convection scheme; with spectral nudging (snON); without spectral nudging (snOFF); surface flux option 0 (sf0); option 1(sf1); and option 2 (sf2). X-axis is**
**the analysis period between 18 UTC 5 November 2013 to 18 UTC 8 November 2013.**

Fig. 5 shows the mean and standard deviation of the biases of the simulated intensities to the choice of the parameterization schemes. There is a statistically significant difference at 99% confidence level (p-value: 0.007941) among the simulations using KF and TK cumulus convection schemes (Fig.5, 1st row). In simulating the intensity, nudging did not demonstrate a
consistent improvement in the intensity of the simulations (Fig. 5, 2nd row). While the choice of surface flux option had a more demonstrable effect on the resulting intensities (at 99% confidence levels) with sf1 having the most intense simulation of the storm in terms of both MSLP and maximum winds and sf0 with least intensity (Fig. 5,3rd row).

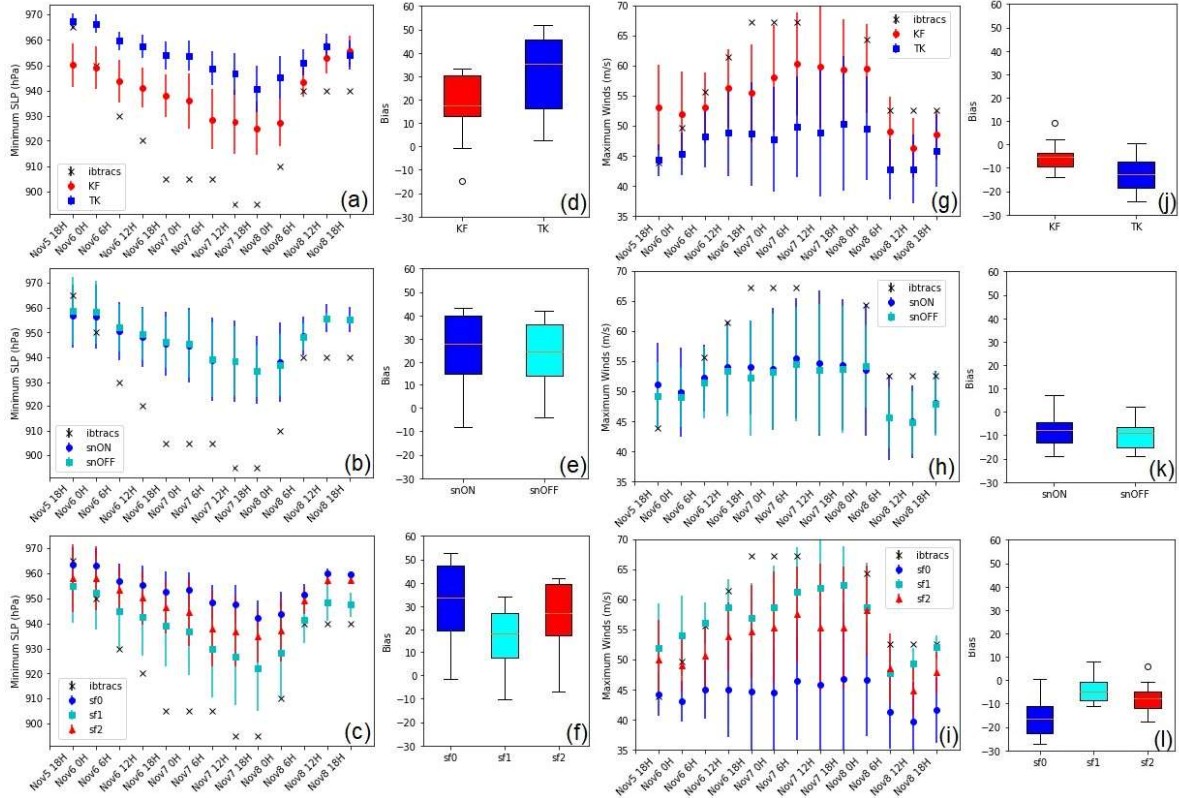


**Figure 5: Time series of the mean intensities and standard deviations (a-c) for MSLP and (g-i) for maximum winds, with mean biases for MSLP (d-f), hPa and maximum winds, m/s for each group (j-l) for cumulus schemes KF and TK, spectral nudging and for surface flux options. X-axis is the analysis period between 18 UTC 5 November 2013 to 18 UTC 8 November 2013.**

Fig. 6 shows that the simulations using the KF scheme have higher correlations and smaller RMSE values than the simulations that used the TK scheme. Of all the simulations, the simulation with the combination of KF and sf1 without nudging has the lowest RMSE (22 hPa MSLP and 9.59 ms$^{-1}$ maximum winds) and highest correlation coefficient of 0.78 and 0.82 for MSLP and maximum winds, respectively. While the simulation with the poorest performance i.e. highest RMSE (37 hPa and 14.17 ms$^{-1}$) and lowest correlation coefficient (0.60 and 0.69 for MSLP and maximum winds, respectively) is the simulation with the 445 combination of TK, sf0, with spectral nudging turned on.

The KF and TK schemes represent shallow convection differently, resulting in different simulated TC intensities (Torn and Davis, 2017). The TK scheme allows both upward transport of moisture across the boundary layer, as well as vertical advection of evaporation from the ocean surface (Parker et al., 2018). Consequently, this reduces the mass flux in deep convection, 450 thereby lowering the rate of TC intensification and resulting in lower simulated intensities. The KF scheme, however, is less likely to reduce the deep convective mass flux that allows for intensification rates to increase. These results are consistent with the differences in the simulated intensities shown in Parker et al. (2017) and Shepherd and Walsh (2016). Parker et al. (2017) found that the KF scheme produces more intense TC systems (lower MSLP values) than the TK scheme for TY Yasi in Australia. Shepherd and Walsh (2016) also found that the KF scheme produces stronger storms (TY Yasi 2011 in the Southwest 455 Pacific and TY Rita 2005 in the north Atlantic) but almost the same intensity for simulations using TK and KF schemes for TY Megi in the Western North Pacific Basin.

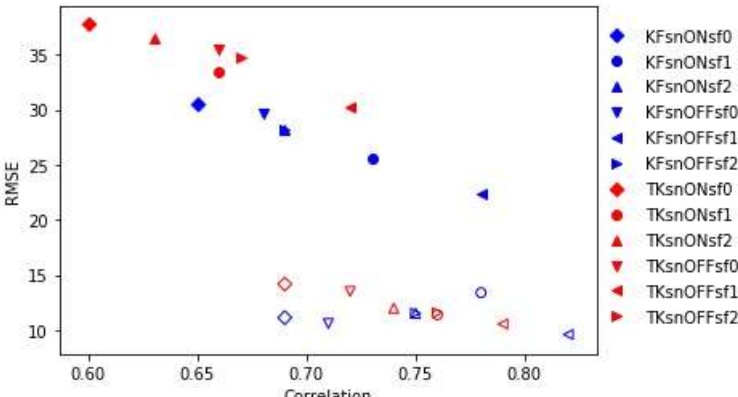

**Figure 6: RMSE vs CC for minimum sea level pressure in hPa (filled) and for maximum winds in ms-1 (not filled) for the sensitivity**
**experiments**

The choice of surface flux option (sf0, sf1, sf2) also affects the ability to reproduce both minimum sea level pressure and maximum winds, as shown by the lower RMSE of sf1 (Fig. 6). Simulations with sf1 have generally shown to have the highest correlation coefficients. While both wind speed and MSLP intensity are strongly dependent on the surface flux option, sf1 465 shows to simulate the highest intensity for TY Haiyan. As in Kueh et al. (2019), the default option (sf0), in which CD does not level off, the simulation of Haiyan has the weakest wind speeds. The sf1 option is expected to have the highest intensity since it has the largest enthalpy and momentum (Ck/CD) ratio at high wind speeds and lowest CD. This gives less friction at high winds, thereby favouring higher intensity (Kueh et al. 2019). The simulated intensity of sf2 option, on the other hand, is expected to be between sf0 and sf1 (Kueh et al. 2019).


Comparing the simulations with the KF and TK schemes shows that the former produces better simulated intensities with lower biases, RMSE and higher correlation coefficients (Fig. 6), consistent with Zhang et al. (2011) and Parker et al. (2017), and minimum sea level pressure as with Spencer and Shaw (2012).

**Table 4: The resulting deviation from landfall location (km, rounded to nearest whole number); translation speed (ms⁻¹, rounded to two decimal place); and deviation from observed translation speed (ms⁻¹, rounded to two decimal places); and deviation of the simulated MSLP at landfall (hPa, rounded to the nearest whole number); compared to observations**

| Simulation | Deviation from landfall point 10.83°N, 125.69°E (in km) | Translation speed before landfall (ms⁻¹) (Obs. 9.48 ms⁻¹) | Deviation from observed translation speed before landfall | Deviation from observed MSLP (895 hPa) at landfall |
|---|---|---|---|---|
| KFsnONsf0 | 56 | 9.62 | 0.14 | 40 |
| KFsnONsf1 | 76 | 8.78 | -0.70 | 18 |
| KFsnONsf2 | 55 | 8.76 | -0.72 | 32 |
| KFsnOFFsf0 | 20 | 9.27 | -0.22 | 46 |
| KFsnOFFsf1 | 37 | 9.54 | 0.05 | 27 |
| KfsnOFFsf2 | 3 | 9.58 | 0.10 | 44 |
| TKsnONsf0 | 6 | 9.80 | 0.32 | 56 |
| TKsnONsf1 | 11 | 9.87 | 0.39 | 43 |
| TKsnONsf2 | 3 | 9.85 | 0.37 | 52 |
| TKsnOFFsf0 | 56 | 9.23 | -0.25 | 49 |
| TKsnOFFf1 | 68 | 9.54 | 0.06 | 38 |
| TKsnOFFsf2 | 61 | 9.20 | -0.29 | 48 |

We also considered the wind-pressure relationship of the simulated intensities of all experiments, which according to Green
and Zhang (2013) is affected by surface flux options. The scatterplot in Fig. 7 indicates the relationship between the MSLP and maximum wind, based on the different simulations. The IBTraCS data (black square markers) are also included in this plot. Almost all simulations show a decreasing trend of the MSLP and maximum winds as the storm intensifies; however, the intensities are evidently underestimated (MSLP and maximum wind speeds). Based on Manganello et al. (2012), the maximum wind speed is usually underestimated in LAMs when the simulated MSLP is below approximately 980 hPa. It is worth pointing
out that of the different simulations, those utilizing the surface flux option 1 (sf1, blue) give the most intense storm by wind speed (Fig. 8). The simulated maximum wind speeds in the simulations using the default surface flux option (sf0, red) only ranges between 35 to 55 ms⁻¹ while the simulations using the other options (sf1, blue and sf2, cyan) are well distributed from ~40 to 73 ms⁻¹, consistent with the result of Kueh et al. (2019). Most simulations have an underestimated maximum wind speeds for MSLP below 910 hPa, which is consistent with a study using WRF that produced lower wind speed compared to
IBTrACS for a given MSLP (Hashimoto et al., 2015).  However, the simulations were able to generate considerable intensity for the maximum wind speed for TY Haiyan compared to that of Islam et al. (2015) who used different model physics options i.e. WRF single moment 6-class (WSM6), WRF single moment 3-class (WSM3), new Thompson (THOM), Milbrandt-Yau double moment (MY2) 7-class scheme, and the Goddard GCE (GGCE) schemes. Previous studies using lower resolution generated insufficient wind speeds in the regime higher than 45 ms⁻¹ (Jin et al., 2015) which are primarily attributed to low
model resolutions and deficiencies in surface drag representations at high wind conditions (Jin et al., 2015; Shen et al., 2017).

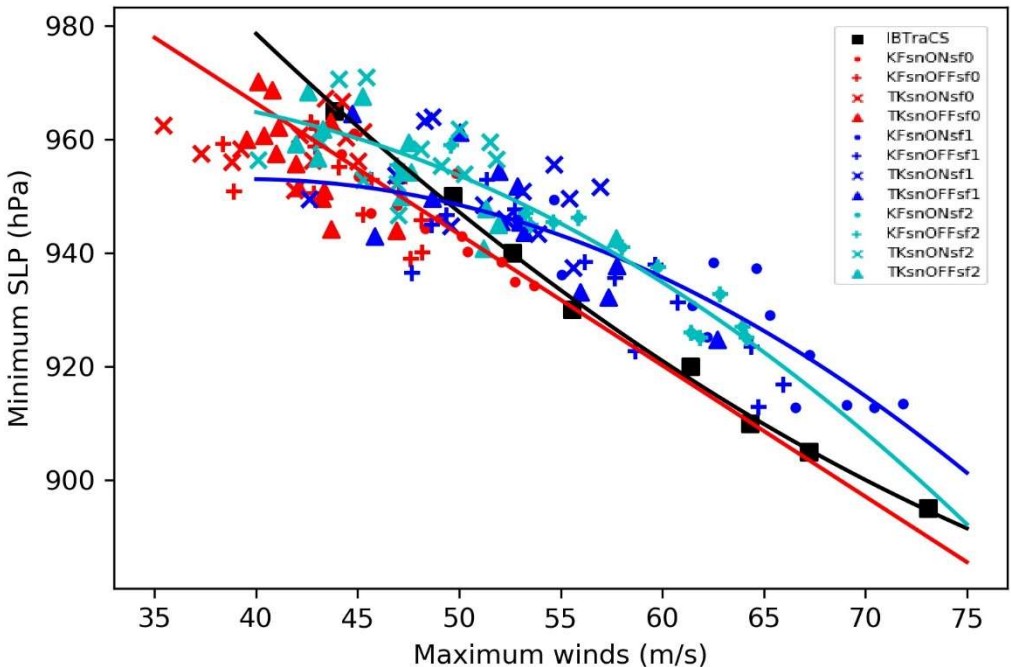

**Figure 7: Scatterplot of MSLP vs maximum wind from the various sensitivity experiments compared with best track data. Solid lines of the corresponding colours (red for sf0, blue for sf1, cyan for sf2) show the second-order polynomial fit.**


In simulating TCs, it is important to get the timing and intensity at landfall right as it gives a good indication of the potential damage along coastal areas (Parker et al. 2017). TY Haiyan made landfall in the eastern-central Philippines (Guiuan, Eastern Samar) on 7 Nov 2013 at 2040 UTC. Figure 11 shows that the simulation with the closest landfall time and location occurs for the KFsnoffsf2 simulation. The deviation from the observed landfall point, minimum deviation is 3 km for KfsnOFFsf2 and

TKsnONsf2 and maximum deviation is 76 km for KFsnONsf1, is within the average forecast error for tropical cyclones at 24-hour lead time in Western North Pacific (Peng et al, 2017).

Fig. 8 also shows that the simulated TY is slightly slower (farther from land on 7 Nov 2013 at 00UTC) than observed, with the timing of landfall delayed between approximately 2 to 6 hours in the simulations. Based on data from IBTrACS, Haiyan's

translation speed before landfall is approximately 9.48 ms$^{-1}$, while the mean translation speed of all the simulations is 9.43 ms$^{-1}$ as shown in Table 4. Fig. 8 also shows that the extent of the wind field of the simulations using KF scheme are wider than the ones using TK scheme. The KF scheme simulations have a bigger radial extent, for winds speeds larger than 35 ms$^{-1}$ or 80 miles per hour (mh$^{-1}$), than the simulations using the TK scheme. The wind field extent is also bigger in simulations with sf1 and sf2 than the ones using the default surface flux option (sf0), with sf1 having a wider and more symmetric radial extent of

winds greater than 50 ms$^{-1}$ or 110 mh$^{-1}$. TY Haiyan's radius of maximum wind was estimated to be between 25-29 km (Shimada et al, 2018). In addition, the radial extent of winds of approximately 15 ms$^{-1}$ (30 mh$^{-1}$) is bigger in simulations using KF than simulations using TK scheme, with radius of maximum wind extending up to ~52 km and ~42 km, respectively. The TKsnONsf0 and TKsnOFFsf0 both have radial extent of winds of 15 ms$^{-1}$ (30 mh$^{-1}$) that are closer to what is estimated using the OSCAT scatterometer data.


### 3.3 Simulated Track and Intensity from ERA5 EDA Ensemble Members

The simulated tracks of TY Haiyan, using the four ERA5 EDA members as initial and boundary conditions and configurations that are the same as used for the control simulation, are found to be within the variability of the simulations using the different

parameterizations (Fig. 9). The average DPE of the ensemble mean is 86 km compared to the average DPE of the simulations using different parameterizations which is 78 km with a range from 7 km to 250 km throughout the whole simulation period. There is no significant difference between the mean DPE of the simulations using the different ensemble members with the simulations using the different parameterization schemes (p-value = 0.464).


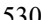

Figure 8: Surface winds (mh⁻¹) (a) from the OSCAT radar scatterometer on the Indian Space Research Organization's OceanSAT-2 satellite at 0130UTC 7 November 2013 and (b-e) for each of the experiments at 00 UTC 7 November 2013. *Source of Figure 8a: https://www.jpl.nasa.gov/images/super-typhoon-haiyan. Use is covered by https://www.jpl.nasa.gov/jpl-image-use-policy*


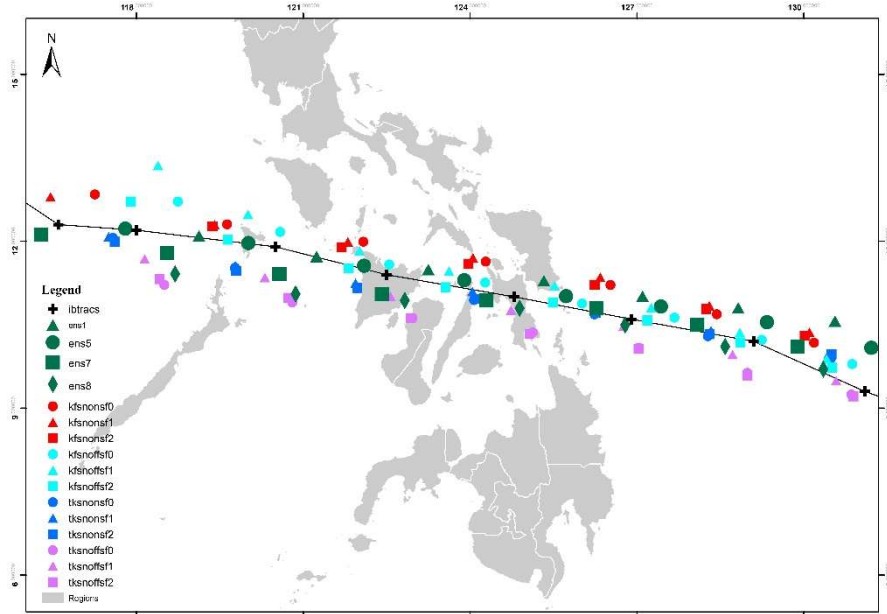

**Figure 9: Simulated tracks of the four randomly selected EDA Ensemble Members (green) compared with IBTRaCS and the sensitivity experiments classified according to experiment groups: Kain-Fristch (KF) convection scheme; TK (TK) convection scheme; with spectral nudging (snON); w nudging (snOFF); surface flux option 0 (sf0); option 1(sf1); and option 2 (sf2).**


The spread in the mean bias of the simulated intensities (MSLP and maximum winds) using the ensemble members as boundary conditions is similar to or within the spread of the correlation between the experiments with the different parameterization schemes and spectral nudging option (Fig. 10). Judging from the spread of the simulate intensities found in the boundary

conditions experiments, the use of different ensemble members has relatively less effect on the simulated intensities as compared to the sensitivity to cumulus and surface flux parameterizations.

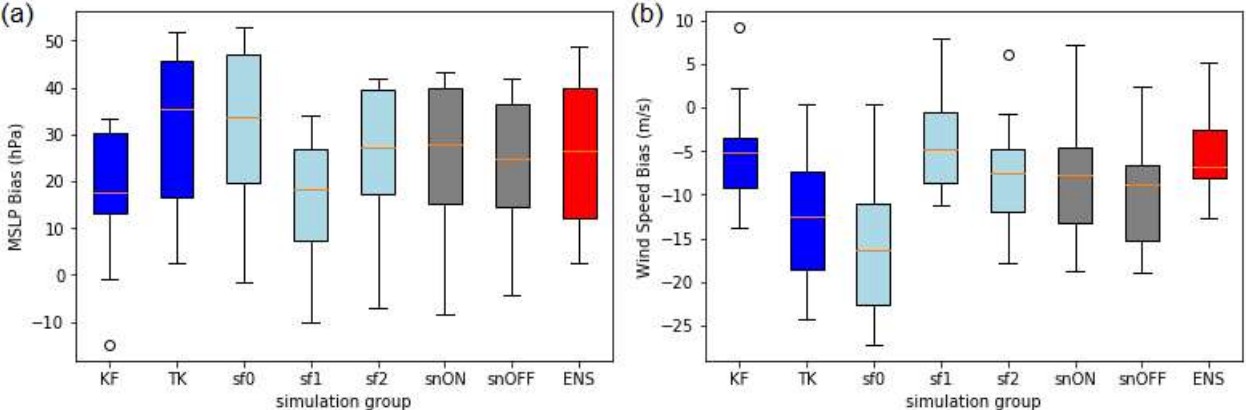

**Figure 10: Mean biases for (a) MSLP in hPa and (b) maximum winds in ms-1 for each group of simulations: cumulus schemes KF and TK (blue bars), surface flux options (sf0, sf1, sf2) (light blue bars), spectral nudging ON and OFF (gray bars), and mean of the different experiments using four randomly selected EDA ensemble members (ENS) (red bars) as initial and boundary conditions.**

### 3.4 Simulated Rainfall


The simulated rainfall in WRF is represented implicitly to demonstrate the effects of sub-grid scale processes through the cumulus scheme and explicitly through the microphysics scheme. In this study, we used the combination of both implicit and

explicit precipitation as the total rainfall. The spatial distribution of rainfall (mm) from 00 UTC 7 November 2013 to 18 UTC 8 November 2013 from the different experiments without spectral nudging are presented in Fig. 11. These results show a discernible difference between the spatial distribution and magnitude of the simulated rainfall, which indicates high sensitivity to the cumulus schemes. The accumulated 6-hourly rainfall was generally larger in magnitude and spatial extent for the simulations using the KF scheme (Fig. 11 b-d) than those that used TK scheme (Fig. 11 e-g). There is not much difference in the magnitude and distribution of rain among the different surface flux options.

It is also important to note the delay in the rainfall at landfall, primarily due to the relatively slower movement of the simulated TCs. The extent of the distribution of rainfall outside of Haiyan's inner rain bands were also not captured well by the simulations when compared with the satellite-derived GPM rainfall (Fig. 11a). In comparison with the GPM rainfall, the distribution of the simulated high rainfall using the KF scheme shows more similar patterns unlike with the TK scheme. The areas of high rainfall appear to be similar in the simulations using different flux options but different in simulations using KF and TK scheme. The simulations using KF scheme also seem to capture the outer rainbands of TY Haiyan, but extending further southeast compared to the GPM rainfall. Previous studies have also indicated the sensitivity of TC-associated rainfall to different physics parameterizations in WRF. Satuya et al. (2019) and Duc et al. (2019) found that KF better predicts rainfall than TK but both generally perform poorly in simulating rainfall, and WRF TC-associated rain are underestimated (Bagtasa 2021).

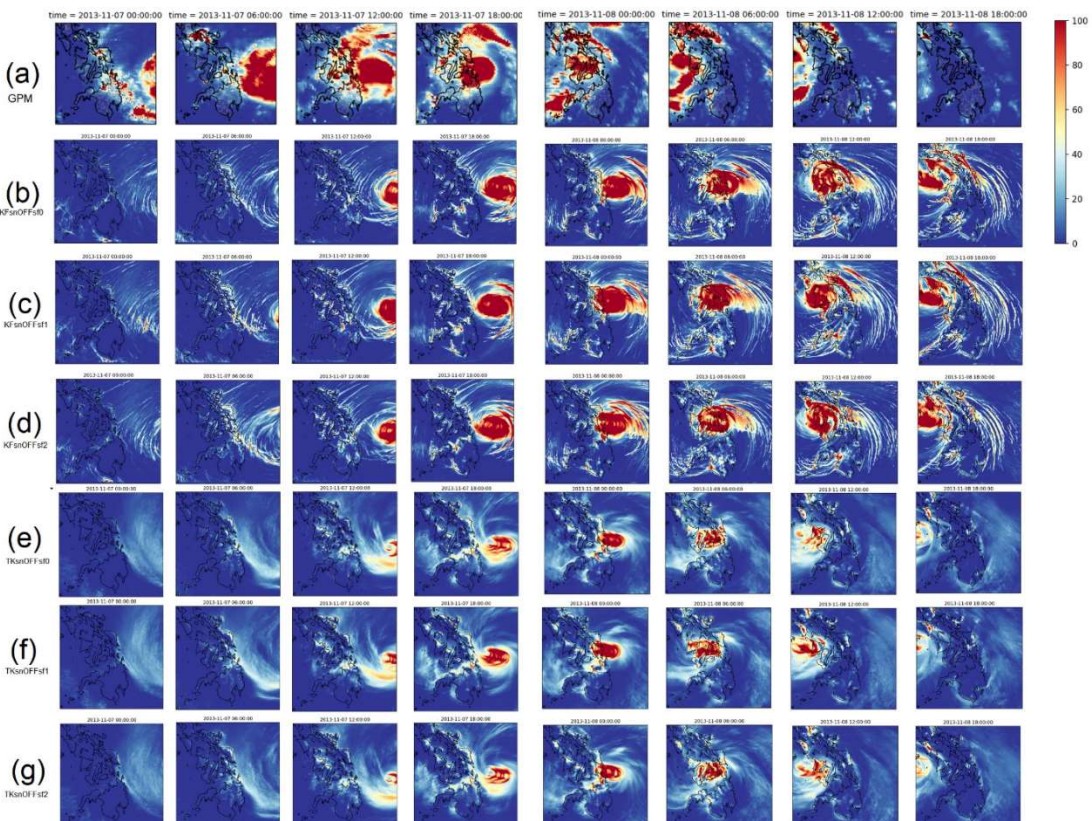

**Figure 11: Spatial patterns of rainfall (in mm) every 6-hours from 00 UTC 7 Nov 2013 to 18 UTC 8 Nov 2013 (a) GPM, and the different simulations without nudging using (b,c,d) KF with sf0, s1,sf2 respectively, and (e,f,g) TK with sf0, sf1, sf2 respectively.**

## 3.5 Environmental Factors

This section discusses the environmental variables to explain the differences between the simulations using KF and TK schemes. KFsnOFFsf1 and TKsnOFFsf1 were used in this section to represent the experiments with KF and TK runs, primarily for improved readability but more importantly, similar results were found in the average of the experiments using KF and TK as cumulus convection scheme. Based on previous similar studies (Parker et al. 2017; Torn and Davis 2012) and as shown in Fig. 12, the KF scheme results in a warm temperature bias (at 700hPa).In particular, the TK scheme produces cooler temperatures and the KF scheme simulates up to approximately 1.5 to 2°C warmer temperatures relative to ERA5, while the ones using the TK scheme have a colder bias at 700hPa (Fig. 12), which is consistent with previous studies (Parker et al. 2017 and Shepherd and Walsh 2016). On the other hand, the KF scheme is likely to simulate the deep convective mass flux which allows for an increase in intensification rates (Zhu and Smith 2002; Emanuel 1989 as cited by Torn and Davis 2012).

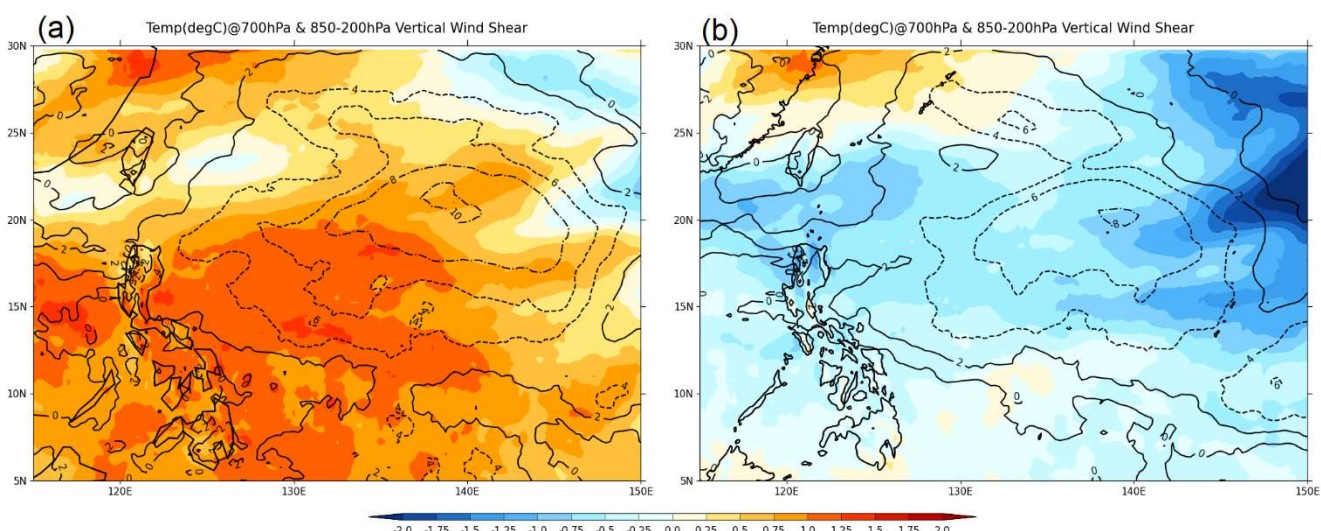

**Figure 12: The difference of the simulated temperature (in degree Celsius) at 700hPa (shaded contours) and deep vertical wind shear (contour lines) averaged over the entire period of the simulation with (a) KF (corresponding to kfsnoffsf1) and (b) TK (corresponding to tksnoffsf1) temperature and winds from ERA5. The 6-hourly WRF output was interpolated to the coarser 6-hourly ERA5 grid using First-order Conservative Remapping through CDO remapcon function. CDO code available at https://code.mpimet.mpg.de/projects/cdo/**

Fig. 12 also displays the simulated deep layer vertical wind shear (contour), which is defined as

$$\text{Vertical Wind Shear} = \sqrt{(U200 - U850)^2 + (V200 - V850)^2}, \qquad (2)$$

where u, v are the zonal and meridional wind components, respectively, at 200 and 850 hPa, computed from time-averaged vertical wind shear calculated from u and v winds at 200 and 850 hPa at each grid-point. The simulated vertical wind shear is weaker along the track of TY Haiyan for both simulations using KF and TK, but the simulation using KF has a bigger area with weaker shear. It is likely that the more homogeneous temperature field in KF resulted in less vertical wind shear, while the simulation using the TK scheme led to a more heterogeneous temperature increasing the vertical shear. A previous study by Floors et al (2011) showed that the temperature differences through the atmospheric profile leads to geostrophic wind shear in WRF simulations. With the weaker vertical shear, the intensity is higher in the simulation using KF than the simulation using TK scheme. Weaker vertical shear has been found to be favourable in maintaining TC development and intensity (Shen et al., 2019).

To further investigate the difference in the track between KF and TK simulation runs, we analysed the 500mb geopotential height. The 5800 m geopotential height contour at 500 mb is used to depict the western North Pacific sub-tropical high (WNPSH) (Xue and Fan 2016). With the ridge location at 20°E, the WNPSH extends to the north of South China Sea (Shen et al., 2019). It has been found that the westward extent and location of the subtropical high ridge directly affect TC tracks in the WNP basin that impact the Philippines (Bagtasa 2020). In the simulation using the KF scheme, the subtropical high is weaker and is substantially in a more northward position compared to the simulation using the TK scheme (Fig. 13), which likely causes the tracks of the simulations using the KF scheme to drift northward, while the simulations using the TK scheme are much closer to the observed. According to Sun et al. (2015), deep convection in mass flux schemes, such as KF, produces large amounts of anvil clouds that warm the upper troposphere and cause latent heating south of the WNPSH that leads to the weakening of the WNPSH and the movement of the TCs northward. Villafuerte et al. (2021) further added that the use of cumulus schemes results in a weaker subtropical high resulting in shifts in northward re-curvature of TC tracks.

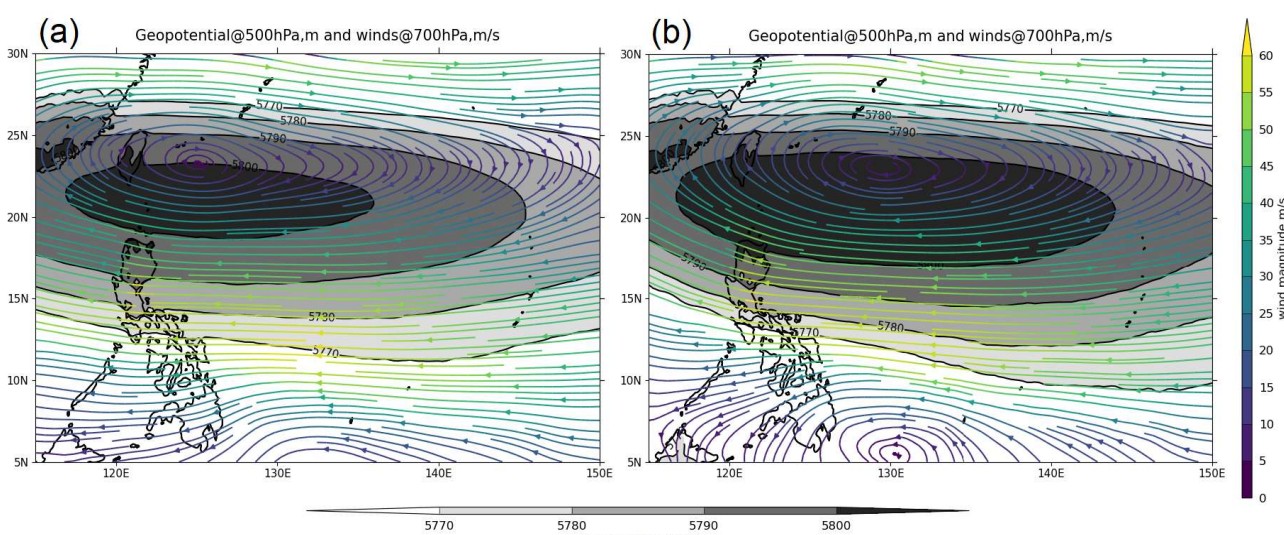

**Figure 13: Geopotential height at 500hPa in geopotential meters shaded contour lines and winds (streamlines) at 700hPa averaged over the entire period of the simulation with (a) KF (corresponding to kfsnoffsf1) and (b) TK (corresponding to tksnoffsf1). The 6-hourly WRF output was interpolated to the coarser 6-hourly ERA5 grid using First-order Conservative Remapping through CDO remapcon function. CDO code available at https://code.mpimet.mpg.de/projects/cdo/**

The TK scheme also produced relatively drier storm environments along the TC path compared to the simulation using KF scheme and as a result, less convection, which translates into weaker intensity (lower wind speeds), whereas simulations using the KF scheme are ~15% higher relative to the simulation using TK. The TK scheme has relatively drier bias with respect to ERA5 along the TC track (Fig. 14). According to Villafuerte et al. (2021), the TK scheme underestimates mid-tropospheric relative humidity, providing a drier environment thereby constraining deep convection and inhibiting TC development. Furthermore, Shen et al. (2019) demonstrated that drier lower troposphere enhances downdrafts and inhibit convection, resulting in weaker intensities and less rain. When comparing the distribution of mid-tropospheric relative humidity as shown in Fig. 14, KF shows a higher relative humidity along the track of Haiyan, which indicates that the KF scheme produces more convection and generates significant rainfall associated with the system, as compared to the weaker convective organization (hence less rainfall) of the simulations using the TK scheme.

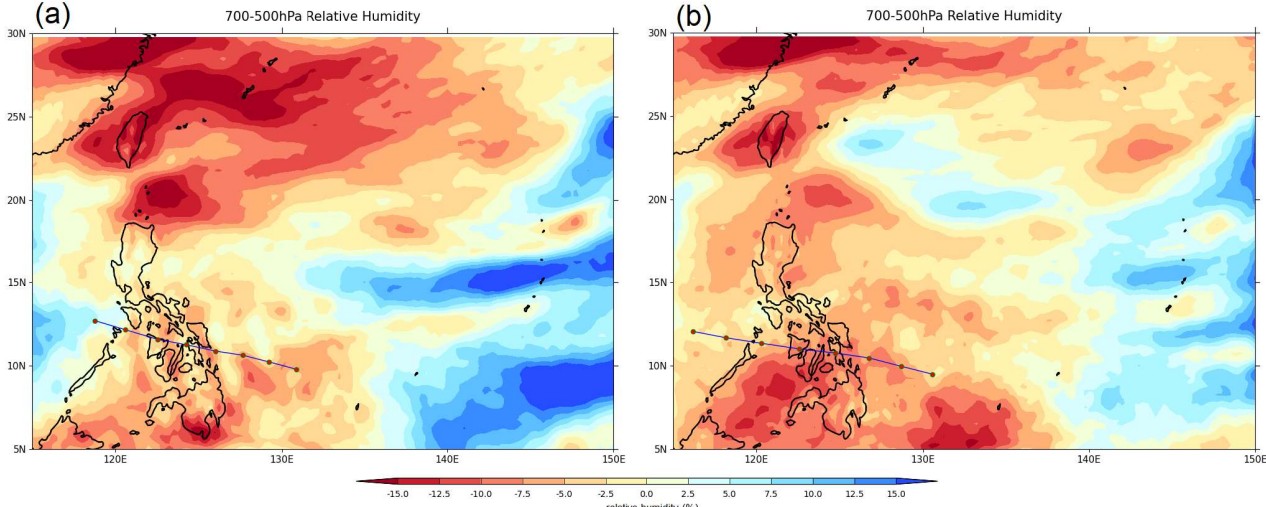

**Figure 14: The difference of the simulated Mid-tropospheric (700-500hPa) Relative Humidity averaged over the entire period of the simulation with (a) KF (corresponding to kfsnoffsf1) and (b) TK (corresponding to tksnoffsf1) from ERA5. The 6-hourly WRF output was interpolated to the coarser 6-hourly ERA5 grid using First-order Conservative Remapping through CDO remapcon function. CDO code available at https://code.mpimet.mpg.de/projects/cdo/**

## 4. Conclusion

Typhoon Haiyan (2013) was one of the most intense and destructive tropical cyclones ever to hit the Philippines. As climate models project more intense storms will occur more frequently in the future due to climate change (e.g. Typhoon Haiyan), it is important to improve their representation in high-resolution models. This will help improve understanding of TCs under

655 climate change and improve confidence in model projections, and more importantly, for risk and impact assessments. The intensity of TY Haiyan proved difficult to simulate using the Weather Research and Forecasting model at 5-km domain configuration as with other previous studies. This study was able to assess the sensitivities to different parameterizations in WRF that can be useful in future simulations of TC cases under future climate conditions. Despite the failure to simulate Haiyan's rapid intensification phase, the simulations were still able to capture the tracks and intensity reasonably well. Based

on the results, there seems to be a trade-off between utilizing KF and TK cumulus schemes that has not been previously discussed in previous studies of tropical cyclones in the Philippines.

The simulated intensity of TY Haiyan is most sensitive to changes in the cumulus scheme and surface flux options; on the other hand, simulated track is most sensitive to cumulus scheme and spectral nudging. However, the TK cumulus scheme

produces better track and the KF scheme produces better intensity. There is a statistically significant difference in the simulated tracks and intensities between the use of the two cumulus schemes. The TK scheme simulates the track better, while the KF scheme produces higher intensities, with the KF scheme simulating a mean bias of 16 hPa and 2 ms$^{-1}$ and the TK scheme with a mean bias of 31 hPa and -6 ms$^{-1}$, respectively. The KF scheme has larger DPEs (mean DPE of $55 \pm 7$ km compared to mean DPE of $47 \pm 5$ km for TK scheme) due to a more northward steering flow. On the other hand, simulations using the TK scheme

had weaker wind and higher MSLP due to the suppression of deep convection by active shallow convection. Simulated rainfall is also sensitive to the cumulus schemes, with simulations using TK having less and smaller rainfall extent than simulations using KF cumulus convection scheme.

The results also show the simulated tracks are sensitive to spectral nudging which results in a reduction in the mean DPE by

675 20 km. The intensity varies as well with different surface flux options. With surface flux option 1, the momentum roughness

length is expressed using a combination of two roughness length formulas (Green and Zhang, 2013), in which the first is Charnock (1955) plus a constant viscous term and the second is the exponential expression from Davis et al. (2008) with a viscous term (as cited by Kueh et al., 2019). Surface flux option 1 simulates better intensities than the other two options (default surface flux option and surface flux option 2). The use of boundary conditions from different ensemble members also resulted in variations in the simulated tracks and intensities, but still within the range of variability of the different parameterization experiments. The use of the KF convective scheme and a more reasonable surface flux option (sf1) can help improve the simulated intensity. While the use of the TK convective scheme and application of spectral nudging can improve the track simulation.

This study is part of an on-going effort to investigate the effect of future climate on the intensity and track of selected destructive TC case studies in the Philippines such as Haiyan using a regional climate model. The resulting sensitivities to the cumulus schemes will be an important consideration in simulating the TC case studies with climate change forcing. Our findings further stress the need for choosing the appropriate cumulus schemes and surface flux parameterization given its impacts on different TC characteristics, e.g. the KF scheme and surface flux option 1 for simulating better intensities of extreme TCs such as Haiyan, besides higher grid resolutions as noted in previous studies (Kueh et al., 2019, Li et al., 2018). The results presented here can also be used in further improving the value of downscaling for simulating intense TCs like Haiyan. These and future results will be useful in addressing the growing need to plan and prepare for, and reduce the impacts of future TCs in the Philippines. As shown in this study, there are uncertainties associated with the use of cumulus parameterizations schemes, spectral nudging and surface flux parameterizations. To cover these uncertainties, the use of ensemble simulations can be applied. For operational applications, an ensemble of cumulus parameterizations can be used to take into account the uncertainty in the track and intensity of simulation intense TCs. This study can facilitate research on regional climate modeling to improve simulations of intense TCs like Haiyan. Furthermore, it is important to study LAMs with a model resolution less than 5 km that can be extremely useful in simulating TCs and associated rain. Li et al. (2018) suggested that a 2-km convection-permitting resolution is needed to reproduce intense TCs such as Haiyan. Other model parameterizations such as cloud microphysics and planetary boundary layer, and ocean coupling may help further improve the intensity simulations of extreme TC such as Haiyan but are beyond the scope of this paper. Simulations using a higher resolution convection-permitting model is needed. Additional simulations and further investigations on these aspects, as well as for other similar TCs will be useful.

**Appendices**

*none*

**Code availability**

Code for the WRF model is available at http://www2.mmm.ucar.edu/wrf/users/downloads.html. WPS Geographical Input data is available from https://www2.mmm.ucar.edu/wrf/users/download/get_sources_wps_geog.html#mandatory.
TRACK is available from https://gitlab.act.reading.ac.uk/track/track. CF-python and CF-plot were used in the analysis and visualization, and installation packages are available from https://ncas-cms.github.io/cf-python/

**Data availability**

Simulation data are stored at the JASMIN data storage facility and is available upon request from the corresponding author.

**Author contribution:**

RJD designed the experiments with guidance from GB, PLV and KH. RJD performed the simulations and analysis with inputs from all co-authors particularly the interpretation of the results. RJD wrote the article with contributions from all co-authors.

**Competing interests:**

The authors declare that they have no conflict of interest.

**Disclaimer**

*none*

**Acknowledgements**

The authors would like to thank the inputs from Dr. Nicholas Klingaman in the design and initial analysis of the experiments. Rafaela Jane Delfino is supported by a scholarship under the Philippine Commission for Higher Education and British Council under the Joint Development of Niche Programme through Philippines-UK Linkages (JDNP) Dual PhD Program.  Kevin Hodges and Pier Luigi Vidale is funded by the RCUK through the Natural Environment Research Council. This research used resources of the JASMIN data analysis facility supported by the Centre for Environmental Data Analysis.  We also thank the
two anonymous reviewers for their constructive comments and suggestions.

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
