# Peer review of "Sensitivity of simulating Typhoon Haiyan (2013) using WRF: the role of cumulus convection, surface flux parameterizations, spectral nudging, and initial and boundary conditions"

_Natural Hazards and Earth System Sciences, 2021_

## Author Comment (AC1)

*Response to comments of **Anonymous Referee #1** on*

**Sensitivity of simulating Typhoon Haiyan (2013) using WRF: the role of cumulus convection, surface flux parameterizations, spectral nudging, and initial and boundary conditions**

Delfino et. al.

**RC1**: 'Comment on nhess-2021-400', Anonymous Referee #1

| Reviewer 1's COMMENTS | Authors' RESPONSES |
|---|---|
| **General comments** | |
| It is an interesting and well-written article that investigates the impact of (a) two different cumulus convection schemes (Kain-Fritsch and Tiedtke), (b) three surface flux formulations, (c) spectral nudging and (d) initial and boundary conditions from ERA deterministic and Ensemble of Data Assimilations system, on the WRF simulations of super Typhoon Haiyan (2013) in Western North Pacific. The model results are compared against the International Best Track Archive for Climate Stewardship, satellite data and ERA5 re-analyses. | Dear Referee,

Thank you very much for the overall positive feedback on the submitted manuscript and for giving us the opportunity to submit an improved version of the manuscript. We appreciate the thoroughness and objectiveness of the comments and have addressed the specific concerns raised. And all changes are highlighted in the revised manuscript. All line numbers refer to the revised manuscript with tracked changes. |
| The use of English is very good. The figures/tables are clearly produced and necessary. The abstract is concise and the conclusions are supported by the results. | Please see below our specific responses and refer to the attached revised manuscript and supplementary file for more details. |
| It is suggested to accept this article for publication after some minor corrections are performed. | |
| **Suggested corrections:** | |
| Section 2.4: (a) Did you use one or two-way nesting? (b) Please justify the location of the southern boundary of the inner domain so close to the track of the tropical cyclone. Errors from the boundary conditions are expected to influence the simulation. (c) Why did you extend the inner domain so much north of the track? Please justify it in the manuscript. Was it necessary in order to simulate appropriately the subtropical ridge? (d) Please clearly state whether all the model results of this article are based on the output of the inner domain. | (a) Two-way nesting was used to allow interaction between the outer and inner domain. This has been indicated in the manuscript (Page 6, Lines 236-239)

(b) Southern boundary – the overall approach of this study is to have a common domain for multiple TC cases in this region (other TC cases not included in this paper, but are the focus of a follow-on paper, about to be submitted) to understand and have a more general set of conclusions on the response of TCs to future warming. We conducted several sensitivity experiments on different domain configurations and specific experiments with adjusted southern boundaries were also conducted (but for a different TC case that |

| | |
|---|---|
| | tracked further south) and it was found that the current domain configuration was optimal in terms of simulated tracks and intensity. Indicated in the manuscript (Page 6, Lines 241-244). *Kindly see Supplementary Figure 1 for more details.* |
| | (c) The northern boundary of the inner domain was also designed to consider multiple TC cases (and for further experiments, not included in this paper) that made landfall to the north of the Philippines and to appropriately simulate the subtropical ridge/Western North Pacific Sub-tropical High and Northeasterly winds. Indicated in Page 6, Lines 241-244 in the revised manuscript. Model results indicated in the manuscript are outputs of the inner domain and this has been indicated in the revised manuscript (Page 6, Line 238) |
| Lines 167-170: How do you explain your result that the simulation with the longer lead-time was the best? | Experiments with different lead times have been conducted prior to the selection of 04 Nov 00 UTC as the initial time (longer lead-time). Other experiments include 04 Nov 06 UTC, 12 UTC, 18 UTC; 05 Nov 00 UTC, 12 UTC; 06 Nov 00UTC, 12 UTC; and Results of these experiments showed that this chosen initial time with longer lead-time is able to simulate the observed track and intensity better than later times. The longer lead-time was used to allow for the simulation of the early stages of development of Typhoon Haiyan, as also used by Nakamura et al. (2016) for Typhoon Haiyan under present-day and future-climate simulations and associated storm surge. The model initialized at 04 Nov 00UTC and 07 Nov 00UTC have simulated tracks closer to observed (IbTRaCS). In addition, when comparing the simulated and observed intensity (minimum sea level pressure and maximum wind speed), it can be noted that in the time series of the simulated intensities, the model takes longer to develop than the observed. This is often seen in regional modeling / limited area modeling, seems to indicate that the model often requires a spin-up period, for example a 36-hour spin-up period was also implemented by Cruz and Narisma 2016 in simulating Tropical Storm Ketsana, to reduce the effect arising from imbalances between the simulated results due to the model physics (microphysics, planetary boundary layer, cumulus) and the initial and boundary conditions (Chu et al., 2018). This behavior was found to be related to the planetary boundary and surface layer parameterizations in WRF (Maldonado et al., 2020) and the time needed for initialization can also be affected by the size of the domain and terrain conditions (Chu et al., 2018). Thus, |

| | we considered the time between 04 Nov 00UTC to 05 Nov 12 UTC as the spin-up period (first 36 hours of simulation and at this period Haiyan was observed to be just developing from a tropical depression to a tropical storm) and the results presented in the manuscript covers the analysis period between 5 Nov 18 UTC to 8 Nov 18 UTC to cover Haiyan's mature stage. We have added a few lines in the revised manuscript (Page 7, Lines 253-255). *Kindly refer to Supplementary Figure 2 for more details.* |
|---|---|
| Line 182: Was the cumulus convection scheme employed in both domains? Please state it clearly. | Yes, we have used the cumulus schemes in both 25km outer and 5km inner domain. We have indicated this in the revised manuscript (Page 7, Line 268). |
| Lines 289-290 and 297-298: the mean DPE of KF simulations is not the same in the former and latter lines. The same happens for the TK simulations. Please make the necessary corrections and update lines 562-563 accordingly. | Thank you for pointing this out. The indicated DPEs in Lines 289-290 (Page 11, Lines 390-391 in revised manuscript) were the mean throughout the simulation period and not the analysis period. We have removed this line and retained the correct figures in Page 11, Lines 390-391 and in the conclusion (Page 24, Lines 687-688). |
| Figure 3, x-axes: is it the simulation time or the 72-hour verification time (as it was stated in line 171)? | Apologies for the confusion. We have revised the time axis of Figures 3, 4 & 5 to reflect the analysis period between 18 UTC 5 November 2013 to 18 UTC 8 November 2013. All experiments were initialized at 00 UTC 4 November 2013. |
| Line 319: in Figure 4 the control simulation (KFsnOFFsf0) has a minimum mslp of about 940 hPa (not 934 hPa) and maximum wind speed less than 50 hPa (not 53.69 m/s). | Thank you again for pointing this out. Same with the issue on the DPE and we have indicated the correct figures in the revised manuscript (Page 12, Lines 422-423). It now reads:

*"The control simulation (denoted as KFsnOFFsf0) has a MSLP value of only 939 hPa and maximum wind speed of 43.47 meters per second (ms-1). Compared to the minimum central pressure of 895hPa and 73 ms-11-min sustained wind speed in the observations, this is a difference of 38 hPa and 29.53ms-1, respectively."* |
| Figure 4: For consistency with the symbols of the other experiments, it is suggested to change the pattern of TKsnOFFsf1 to dotted line. In the current figure it is difficult to distinguish it from TKsnOFFsf0. | As suggested, we have revised the figures for better representation of the different experiments. Kindly refer to the updated Figure 4 (Page 13) in the revised manuscript. |
| Lines 349-350: in figure 6 the RMSE of KFsnOFFsf1 is about 10 m/s and its correlation is between 0.8 and 0.85 (i.e. lower than 0.89). | Thank you for spotting this. We have indicated the correct figures in the revised manuscript (Page 15, Lines 458-459). It now reads:
*"Of all the simulations, the simulation with the combination of KF and sf1 without nudging have the lowest RMSE (22 hPa MSLP and 9.59 ms$^{-1}$ maximum winds) and highest correlation coefficient of 0.78 and 0.82 for MSLP and maximum winds, respectively."* |

| | |
|---|---|
| Lines 351-352: in figure 6 the RMSE of TKsnONsf0 is about 15 m/s and its correlation is about 0.69. | Thank you for spotting this. We have indicated the correct figures in the revised manuscript (Page 15, Line 459-461). It now reads:

 *"While the simulation with the poorest performance i.e. highest RMSE (37 hPa and 14.17 ms$^{-1}$) and lowest correlation coefficient (0.60 and 0.69 for MSLP and maximum winds, respectively) is the simulation with the combination of TK, sf0, with spectral nudging turned on."* |
| Line 409: The simulation with the closest landfall time is not shown in Table 3, but it can be derived by Figure 11 (as far as the experiments without spectral nudging are concerned). | Thank you for spotting this. We have indicated the correct figures in the revised manuscript (Page 17, Line 518). |
| Line 464: Please justify your choice to present only the runs without nudging in figure 11. | For improved readability, we have chosen to present the experiments without nudging to represent the TC-associated rainfall in the different experiments. Similar rainfall patterns were found in the experiments with nudging as shown in Supplementary Figure 4. |
| Line 488: the steering flow bias has not been shown in figure 12. | Thank you for pointing this out. We have removed this line in the manuscript (Page 21, Line 600). |
| Figures 12 and 14: Did you interpolate the WRF output to the coarser ERA5 grid? Which interpolation method did you use? Please include this information in the article. | Yes, the 6-hourly WRF output was interpolated to the coarser ERA5 grid using First-order Conservative Remapping through CDO's remapcon function. We have specified this in the revised manuscript (Page 21, Lines 611-613; Page 22, 645-647; Page 23, Lines 665-667) |
| Figures 12, 13, 14: (a) Please justify the use of the KFsnOFFsf1 and TKsnOFFsf1 experiments instead of all the KF and TK runs. (b) are these figures based on 6-hourly ERA5 and WRF output? | (a) KFsnOFFsf1 and TKsnOFFsf1 were used in this section to represent the experiments with KF and TK runs, primarily to save on space but more importantly, similar results were found in the average of the experiments using KF and TK as cumulus convection scheme. *Kindly refer to Supplementary Figures 5,6 and 7.*
 (b) Yes, these are based on 6-hourly ERA5 and WRF output values. We have indicated this in the revised manuscript (Page 21, Lines 611-613; Page 22, 645-647; Page 23, Lines 665-667) |
| Lines 501-502: Please clarify in the article whether the vertical wind shear was computed (a) from time-averaged u and v winds at 200 and 850 hPa (i.e. firstly calculating the time-averaged u and v at each grid-point and then using them to calculate the vertical wind shear), or (b) by averaging the instantaneous values of the vertical wind shear (i.e. firstly calculating the | The vertical wind shear was re-computed by averaging the instantaneous values of the vertical wind shear (i.e. firstly calculating the instantaneous vertical wind shear at each grid-point and then calculating its time-average value) (kindly see revised Figure 12). We specified this in the revised manuscript (Page 22, Lines 619-620) |

| | |
|---|---|
| instantaneous vertical wind shear at each grid-point and then calculating its time-average value). | |
| Line 536: (a) do you mean that KF shows a higher relative humidity along the track? Otherwise, it disagrees with the previous discussion in this paragraph. (b) for clarity it is suggested to draw the tracks of the simulated and actual tracks on both panels of figure 14. | (a) Thank you for pointing this out. We have corrected this observation in the manuscript (Page 23, Lines 657-658). (b) We think this is a great suggestion so we have revised the figure to show the simulated tracks. Please refer to the new Figure 12 in the revised manuscript. |
| **Technical corrections:** | |
| Line 152: "… and model physics (Isaksen et al., 2010)." | Revised in the manuscript (Page 6, Line 230) |
| Line 158: "… different parameterization …" | Revised in the manuscript (Page 6, Line 239) |
| Line 165: It is a 180-hour period (not 174-hour) from 00 UTC 4 November to 12 UTC 11 November. | Revised in the manuscript (Page 6, Line 248) |
| Line 175: "… is bounded by 100-170 degrees East …" | Revised in the manuscript (Page 7, Line 260) |
| Line 251: "… maximum 10m winds to evaluate …" | Revised in the manuscript (Page 10, Line 352) |
| Line 268: "… relative vorticity maxima …" | Revised in the manuscript (Page 10, Lines 368-369) |
| Line 286: "… without nudging (snOFF) …" | Revised in the manuscript (Page 11, Line 387) |
| Line 312: "… of the DPE (km) …" | Revised in the manuscript (Page 12, Line 414) |
| Line 473: "… the KF scheme shows …" | Revised in the manuscript (Page 20, Line 582) |
| Lines 496, 527, 543: KFsnOFFsd1 and TKsnOFFsd1 must be corrected to KFsnOFFsf1 and TKsnOFFsf1, respectively. | Revised in the manuscript (Lines 610, 645, 665) |

**References**

Chu, Q., Xu, Z., Chen, Y. and Han, D.W.: Evaluation of the ability of the Weather Research and Forecasting model to reproduce a sub-daily extreme rainfall event in Beijing, China using different domain configurations and spin-up times. Hydrology and Earth System Sciences Discussions, 22, 3391–3407. https://doi.org/10.5194/hess-22-3391-2018, 2018.

Cruz, F. and Narisma, G.: WRF simulation of the heavy rainfall over Metropolitan Manila, Philippines during tropical cyclone Ketsana: a sensitivity study. Meteorology and Atmospheric Physics, 128(4), 415–428. https://doi:10.1007/s00703-015-0425-x, 2016.

Maldonado T, Amador JA, Rivera ER, Hidalgo HG, Alfaro EJ.: Examination of WRF-ARW Experiments Using Different Planetary Boundary Layer Parameterizations to Study the Rapid Intensification and Trajectory of Hurricane Otto (2016). Atmosphere, 11(12):1317. https://doi.org/10.3390/atmos11121317, 2020.

Nakamura, R., Shibayama, T., Esteban, M. and Iwamoto, T.: Future typhoon and storm surges under different global warming scenarios: case study of typhoon Haiyan (2013). Nat. Hazards, 82, 1645-1681. https//doi:10.1007/s11069-016-2259-3, 2016.

---

## Author Comment (AC2)

*Supplement of the response to comments of Reviewers on*

**Sensitivity of simulating Typhoon Haiyan (2013) using WRF: the role of cumulus convection, surface flux parameterizations, spectral nudging, and initial and boundary conditions**

Delfino et. al.

This supplement contains figures to support responses to comments of Anonymous Referee #1&2.

The overall approach of this study is to have a common domain for multiple TC cases in this region (other TC cases not included in this paper, but are the focus of a follow-on paper, about to be submitted) to understand and have a more general set of conclusions on the response of TCs to future warming. Initial simulations have been done to check model performance using different domain configurations and horizontal resolution i.e. (a) single domain (at 12km horizontal resolution); (b) two domains (at 12 and 4km horizontal resolution); (c) same as (b) but with bigger inner domain; (d) three domains (12, 4 and 1.3km horizontal resolution); and (e) two domains (25,5km) horizontal resolution. Domain configuration (e) was used for the sensitivity experiments which simulated the lowest minimum sea level pressure and maximum winds, and in consideration of computing resources and other TC cases that were simulated in the project.

[Figure]

**Supplementary Figure 1a: Different domain set-up (a-e) for experiments looking at different domain configurations for Typhoon Haiyan with the corresponding simulated minimum sea level pressure (f) and maximum winds (g) for each domain set-up.**

We also conducted several sensitivity experiments on different domain configurations and specific experiments with adjusted southern boundaries were also conducted (but for a different TC case that tracked further south) and it was found that the current domain configuration was optimal in terms of simulated tracks and intensity.

[Figure]

**Supplementary Figure 2b: Different domain set-up (a1, a2, a3), corresponding simulated tracks (b1,b2,b3), simulated minimum sea level pressure (c) and maximum winds (d) for experiments looking at the impacts of the southern boundary for a TC case (Washi, December 2011) that tracked south of Haiyan.**

Experiments with different lead times have been conducted prior to the selection of 04 Nov 00 UTC as the initial time (longer

40 lead-time). Other experiments include 04 Nov 06, 12, 18 UTC; 05 Nov 00, 12 UTC; 06 Nov 00, 12 UTC; and results of these experiments showed that this chosen initial time with longer lead-time is able to simulate the observed track and intensity better than later times.

[Figure]

45

**Supplementary Figure 3: Time series of (a) minimum sea level pressure in hPa and (b) maximum winds in ms-1 for the sensitivity experiments with different initial times, including the simulated tracks (c) for the experiments initialized at 04 Nov 00UTC, 05 Nov 00UTC, 06 Nov 00UTC, and 07 Nov 00UTC.**

50 There is no difference in the simulated intensity (MSLP = 1005hPa; max winds = 17 m/s) at t=0 (04 Nov 00 UTC) for both mother/outer domain (D01) and child/inner domain (D02) for all sensitivity experiments and small differences up to t=12.

[Figure]

**Supplementary Figure 4: Time series of simulated 6-hourly (a) minimum sea level pressure in hPa and (b) maximum winds in ms-1 for the sensitivity experiments from 04 Nov 00 UTC (t=0) to 11 Nov 18 UTC (t=186) from the mother/outer domain (D01) and child/inner domain (D02) for all sensitivity experiments.**

[Figure]

**Supplementary Figure 5: Spatial patterns of rainfall (in mm) every 6-hours from 00 UTC 7 Nov 2013 to 18 UTC 8 Nov 2013 (a) GPM, and the different simulations WITH nudging using (b,c,d) KF with sf0, s1,sf2 respectively, and (e,f,g) TK with sf0, sf1, sf2 respectively.**

[Figure]

**Supplementary Figure 6: The average difference of the simulated temperature (in degree Celsius) at 700hPa (contour) and deep vertical wind shear averaged over the entire period of the simulation with (a) KF and (b) TK temperature and winds from ERA5. The 6-hourly WRF output was interpolated to the coarser 6-hourly ERA5 grid using First-order Conservative Remapping through CDO remapcon function. CDO code available at https://code.mpimet.mpg.de/projects/cdo/**

[Figure]

**Supplementary Figure 7: Average Geopotential height at 500hPa in geopotential meters (shaded contour lines) and winds (streamlines) at 700hPa averaged over the entire period of the simulation with (a) KF and (b) TK.**

[Figure]

**Supplementary Figure 8: The average difference of the simulated Mid-tropospheric (700-500hPa) Relative Humidity averaged over the entire period of the simulation with (a) KF and (b) TK from ERA5.**

---

## Author Comment (AC3)

*Response to comments of **Anonymous Referee #2** on*

**Sensitivity of simulating Typhoon Haiyan (2013) using WRF: the role of cumulus convection, surface flux parameterizations, spectral nudging, and initial and boundary conditions**

Delfino et. al.

**RC2**: ' https://https://doi.org/10.5194/nhess-2021-400-RC2', **Anonymous Referee #2**
Recommendation: **MAJOR REVISION**

| Reviewer 2's COMMENTS | Authors' RESPONSES |
|---|---|
| **General comments** | |
| The authors utilized WRF-ARW to simulate Typhoon Haiyan and investigate the role of cumulus convection (KF and TK schemes), surface flux parameterizations, spectral nudging, and initial and boundary conditions (ERA5 and EDA). They concluded that the TK scheme and spectral nudging improve track simulations with lower mean DPE than the other model configurations. On the other hand, KF scheme and varying the surface flux options improve the intensity. | Dear Referee, Thank you very much for highlighting the importance of our work, the useful feedback on the submitted manuscript, and for giving us the opportunity to submit a much improved version of the manuscript. We have addressed the major and minor concerns raised. All changes are highlighted in the revised manuscript and line numbers refer to the revised manuscript with tracked changes. |
| This type of study will definitely be of a great addition to works that optimize a model's configuration of TC simulations in the Philippines, but in its current form is not yet ready for publication. Major parts of the paper should be rewritten due to the following major concerns: | Please see below our specific responses and refer to the attached revised manuscript and supplementary file for more details. |
| **Major Concerns:** | |
| 1. (Line 55~Line 105, Line 125…) Although a future plan for conducting pseudo-global warming simulations was mentioned, WRF-ARW was used in the paper as a numerical weather prediction (NWP) model to simulate a weather event (TC Haiyan). However, the literature review (introduction) seems to interchange regional climate models (climatological simulations) with numerical weather prediction models (short-term weather events) resulting in mixed and improper | Thank you for pointing this out. The overall approach of the study is that we have used WRF configured as NWP to get the best configuration for hindcast TC case simulations and eventually use that configuration to simulate the TC cases with future climate forcings. The results included in this paper are from the former i.e. as a sensitivity study using Typhoon Haiyan as the TC case. We have revised the manuscript to make the distinction clearer i.e. studies with NWP event-based hindcast simulations to build a foundation on sensitivities to |

| | |
|---|---|
| citations of papers that use RCMs and NWPs. Event simulations are different from climatological runs. Although WRF and other NWPs can also be used as RCM, they are usually modified to efficiently work for climatological simulations (e.g. CLWRF, RegCM -- RCM version of MM5, NHRCM – RCM version of JMA/MRI NHM). NHRCM, and not WRF, is the model used by Cruz et al., 2016 in Line 132. | model parameterizations and settings. We have also cited some studies using WRF as LAM with future climate forcings as initial and lateral boundary conditions in support of the rationale behind the bigger study. Significant revisions were made in Pages 2-4, Lines 55 – 170 in the revised manuscript.

Apologies for this mistake. Cruz et al 2016 should read Cruz and Narisma 2016. We have revised this in Page 1, Lines 71-72 and included in references of the revised manuscript. |
| The paper literature review should focus on studies that conduct TC short-term simulations using models (e.g. WRF, NHM) that are considered as NWP and not RCM.

The literature review also fell short in terms of discussing studies that tackle the other sensitivity parameters such as spectral nudging, surface flux, and ICBC. The reviewer hopes to see a clearer revised Introduction with an additional review on the said parameters. | We have included additional discussion in the introduction, particularly that of surface flux options e.g. from a study by Kueh et al., 2019 using WRF (Page 3, Lines 88-102 of the revised manuscript). Additional studies on ICBC (Islam et al., 2015; Mohanty et al., 2010; Shepherd and Walsh, 2016) and spectral nudging (in WRF as NWP Mori et al., 2014; Kueh et al., 2019 and as RCM Shen et al., 2017; Cha et al., 2011) have also been added in the introduction section (Pages 2-3, Lines 88-123). |
| **2.** The objective and analysis of this paper are very promising but the initial forcing is also very critical to consider it as a sensitivity analysis. Kindly clarify if the researchers downscaled only one mother domain (D1) for all D2 sensitivity runs? If not, then it will be inappropriate and difficult to compare the sensitivity of TC track and intensity to parameterizations if the initial forcing (D1) for each experiment have different model physics. This might explain the different (or larger differences of) values of intensities at t=0 in Figure 4. The reviewer strongly suggests to reconsider rerunning all simulations using only one D1 simulation as forcing to all D2 experiments. | Thank you for these clarifications. There is only one mother/outer domain (D01) and child/inner (D02) domain and the same domain settings were used in all the sensitivity experiments (as shown in Figure 1 of the submitted manuscript). The same physics parameterizations were also used in both outer (D01) and inner (D02) domains. We have explicitly indicated these in the text (Page 6, Lines 235-245) and in Table 3 of the revised manuscript.

Since we are using two-way nesting and there is feedback from the outer to the inner domain and vice versa, it is important that the same physics parameterization is used in both domains. This is the used in WRF with multiple and nested domains (Werner and Wang, 2017; Dudhia 2015), as there could be issues with two-way nesting when physics parameterization differs across the nest boundaries (e.g. in precipitation fields of the mother/outer domain) (Dudhia 2015) and used in past studies (e.g. Wang and Wang, 2014; Islam et al., 2015). The physics parameterization, particularly the cumulus scheme, was changed in each sensitivity experiment in both domains.

Apologies for the confusion. We have revised the time axis of Figures 3, 4 & 5 to reflect the analysis period between 18 UTC 5 November 2013 to 18 UTC 8 November 2013. All |

| | experiments were initialized at 00 UTC 4 November 2013 (t=0). The different values of intensities at the start of the analysis period (18 UTC 5 November 2013) is expected since there has already been interaction between D01 & D02. The same initial conditions were used for D01 and D02. There is no difference in the simulated intensity (MSLP = 1005hPa; max winds = 17 m/s) at t=0 (04 Nov 00 UTC) for both mother/outer domain (D01) and child/inner domain (D02) for all sensitivity experiment. Kindly refer to Supplementary Figure 3 for more details.

 Given this clarification, there is no need to rerun the simulations. |
|---|---|
| With this 2nd major concern, it will be difficult to give meaningful comments on the results and discussions. | Given what we have explained above, there is no reason for the 2nd concern. |
| **3. (Line 155-163, 166).** Kindly provide supplementary materials for the results of the other domain configurations that led the authors to select the control run model setup. These supplementary materials are very important to justify the model setup of the control run. | Thank you for this suggestion. We have included some figures in the supplementary material. Initial simulations have been done to check model performance using different domain configurations and horizontal resolution i.e. (a) single domain (at 12km horizontal resolution); (b) two domains (at 12 and 4km horizontal resolution); (c) same as (b) but with bigger inner domain; (d) three domains (12, 4 and 1.3km horizontal resolution); and (e) two domains (25, 5km) horizontal resolution. Domain configuration (e) was used for the sensitivity experiments which simulated the lowest minimum sea level pressure and maximum winds, and in consideration of computing resources and other TC cases that were simulated in the project. Kindly refer to Supplementary Figure 1a for more details.

 Experiments with different lead times have also been conducted prior to the selection of 04 Nov 00 UTC as the initial time (longer lead-time) as well as experiments on different domain configurations and specific experiments with adjusted southern boundaries were also conducted (but for a different TC case that tracked further south) Kindly refer to Supplementary Figures 1b and 2 for more details.

 For the choice of cumulus parameterization in the control run, we have chosen KF for the control run since it's used by PAGASA in its NWP configuration; the default surface flux option (isftcflx = 0) and no spectral nudging so that we can easily assess the sensitivity to these physics parameterization |

| | and alternative model options. Other parameterizations were based on previous work on Typhoon Haiyan i.e. Li et al., 2018. |
|---|---|
| **Minor suggestions** | |
| **(Line 113):** Correct the year "2012" to "2013". | Thank you for spotting this. Revised in Page 5, Line 190 in the revised manuscript |
| **(Line 125):** Kindly reconsider "NWP" instead of "RCM". | The overall approach of the study is that we have used WRF configured as NWP to get the best configuration for hindcast TC cases simulations and eventually use that configuration to simulate the TC cases with future climate forcings. The results included in this paper is from the former i.e. as a sensitivity study using Typhoon Haiyan as the TC case. We have revised the manuscript to indicate that we used WRF as a LAM so as to avoid confusion. |
| There is no "Powers 2016" in the references. | Apologies for this. Powers 2016 should read Powers 2017. We have revised in Page 5, Line 204 in the revised manuscript and already indicated in the references. |
| **(Line 132):** Cruz et al., 2016 uses NHRCM and not WRF to make temperature and rainfall projections in the Philippines. | Apologies for this mistake. Cruz et al 2016 should read Cruz and Narisma 2016. We have revised this in Page 4, Line 210 and included in references of the revised manuscript. |
| **(Line 155-170):** Kindly provide a table for your control run's model setup as indicated in this section. Make sure to clarify if you performed one-way or two-way nesting, specify the input forcing, temporal and spatial resolutions (dt,dx,dy,dz), model physics, and so on. | We have used two-way nesting (between the outer domain D01 and inner domain D02) with horizontal resolution of 25km for D01 and 5km for D01; and 44 vertical levels with model top of 50hPa. We have explicitly indicated this in the manuscript and added a table for easier reference. Please refer to Table 3, Pages 9-10 of the revised manuscript. |
| **(Line 180):** "These cumulus schemes are used because PAGASA uses KF …". Does PAGASA also uses TK? Does the writer mean "The KF cumulus scheme was used because …"? | PAGASA uses KF, and TK is used for tropical ocean applications. We have indicated this in Page 7, Lines 266-273 in the revised manuscript. |
| **(Line 185):** There is no Sun et al., 2019 in the references. | Thank you for spotting this. Should read and have added Sun et al., 2015 in the text and references. |
| The discussion on TK is too short and vague. The author should also provide short discussion of the main output of the cited references. Same comment for Lines 194-195, 205. | We have revised and added in the discussion on cumulus parameterization particularly on Tiedtke scheme, and added a brief description on the outputs of the cited references. Kindly see Page 7, Lines 274-284. |
| **(Line 206):** Check repeating phrases in the sentence with "Charnock's (1995)". | Thank you for pointing this out. We have revised this in Page 8, Line 303. |

**References**

Cha, D-H., Jin, C-S., Lee, D-K., Kuo, Y-H.: Impact of intermittent spectral nudging on regional climate simulation using Weather Research and Forecasting model. Journal of Geophysical Research: Atmospheres. https://https://doi.org/10.1029/2010JD015069, 2011.

Cruz, F. and Narisma, G.: WRF simulation of the heavy rainfall over Metropolitan Manila, Philippines during tropical cyclone Ketsana: a sensitivity study. Meteorology and Atmospheric Physics, 128(4), 415–428. https://doi:10.1007/s00703-015-0425-x, 2016.

Dudhia, J.: Overview of WRF Physics. Retrieved from http://homepages.see.leeds.ac.uk/~lecag/wiser/sample_wiser_files.dir/Physics_Dudhia.ppt.pdf 2015.

Li, F., Song, J. and Li, X.: A preliminary evaluation of the necessity of using a cumulus parameterization scheme in high-resolution simulations of Typhoon Haiyan (2013). Nat Hazards 92, 647–67. https://doi:10.1007/s11069-018-3218-y, 2018.

Kueh, M.-T., Chen, W.-M., Sheng, Y.-F., Lin, S. C., Wu, T.-R., Yen, E., Tsai, Y.-L., and Lin, C.-Y.: Effects of horizontal resolution and air–sea flux parameterization on the intensity and structure of simulated Typhoon Haiyan (2013), Nat. Hazards Earth Syst. Sci., 19, 1509–1539, https://doi:10.5194/nhess-19-1509-2019, 2019.

Islam, T., Srivastava, P.K., Rico-Ramirez, M.A. et al.: Tracking a tropical cyclone through WRF–ARW simulation and sensitivity of model physics. Nat Hazards 76, 1473–1495, https://doi:10.1007/s11069-014-1494-8, 2015.

Mohanty, U. C., Osuri, K. K., Routray, A., Mohapatra, M., and Pattanayak, S.: Simulation of bay of bengal tropical cyclones with wrf model: Impact of initial and boundary conditions. Marine Geodesy, 33(4), 294–314. https://doi:10.1080/01490419.2010.518061, 2010.

Mori, N., Kato, M., Kim, S., Mase, H., Shibutani, Y., Takemi, T., Tsuboki, K., and Yasuda, T.: Local amplification of storm surge by Super Typhoon Haiyan in Leyte Gulf. Geophysical Research Letters, Volume 41, Issue 14, Pages 5106-5113. https://doi:10.1002/2014GL060689, 2014.

Shen, W., Tang, J., Wang, Y., Wang, S., and Niu, X.: Evaluation of WRF model simulations of tropical cyclones in the western North Pacific over the CORDEX East Asia domain. Climate Dynamics, 48(7), 2419–2435. https://doi:10.1007/s00382-016-3213-5, 2017.

Shepherd, T. J., and Walsh, K. J.: Sensitivity of hurricane track to cumulus parameterization schemes in the WRF model for three intense tropical cyclones: impact of convective asymmetry. Meteorog. Atmos. Phys., 129(4), 345–374. https://doi:10.1007/s00703-016-0472-y, 2017.

Werner, K. and Wang, G: Nesting in WRF. Climate Science Org Australia. Retrieved from /https://www.climatescience.org.au/sites/default/files/werner_nesting.pdf, 2017.

---

## Author Response (AR1)

**Response to comments of Anonymous Referee 1 & 2 on**

**Sensitivity of simulating Typhoon Haiyan (2013) using WRF: the role of cumulus convection, surface flux parameterizations, spectral nudging, and initial and boundary conditions**

**Delfino et. al.**

**RC1: 'Comment on nhess-2021-400', Anonymous Referee #1**

| Reviewer 1's COMMENTS                                 | Authors' RESPONSES                                                |
|--------------------------------------------------------------|-------------------------------------------------------------------|
| General comments                                             |                                                                   |
| It is an interesting and well-written article that           | Dear Referee,                                                     |
| investigates the impact of (a) two different cumulus         |                                                                   |
| convection schemes (Kain-Fritsch and Tiedtke), (b)           | Thank you very much for the overall positive feedback on the      |
| three surface flux formulations, (c) spectral nudging        | submitted manuscript and for giving us the opportunity to submit  |
| and (d) initial and boundary conditions from ERA             | an improved version of the manuscript. We appreciate the          |
| deterministic and Ensemble of Data Assimilations             | thoroughness and objectiveness of the comments and have           |
| system, on the WRF simulations of super Typhoon              | addressed the specific concerns raised. And all changes are       |
| Haiyan (2013) in Western North Pacific. The model            | highlighted in the revised manuscript. All line numbers refer to  |
| results are compared against the International Best          | the revised manuscript with tracked changes.                      |
| Track Archive for Climate Stewardship, satellite data        |                                                                   |
| and ERA5 re-analyses.                                        | Please see below our specific responses and refer to the attached |
| The use of English is very good. The figures/tables are      | revised manuscript and supplementary file for more details.       |
| clearly produced and necessary. The abstract is concise      |                                                                   |
| and the conclusions are supported by the results.            |                                                                   |
| It is suggested to accept this article for publication after |                                                                   |
| some minor corrections are performed.                        |                                                                   |
| Suggested corrections:                                       |                                                                   |
| Section 2.4: (a) Did you use one or two-way nesting?         | (a) Two-way nesting was used to allow interaction between the     |
| (b) Please justify the location of the southern boundary     | outer and inner domain. This has been indicated in the            |
| of the inner domain so close to the track of the tropical    | manuscript (Page 6, Lines 236-239)                                |
| cyclone. Errors from the boundary conditions are             | (b) Southern boundary – the overall approach of this study is to  |
| expected to influence the simulation. (c) Why did you        | have a common domain for multiple TC cases in this region         |
| extend the inner domain so much north of the track?          | (other TC cases not included in this paper, but are the focus     |
| Please justify it in the manuscript. Was it necessary in     | of a follow-on paper, about to be submitted) to understand        |
| order to simulate appropriately the subtropical ridge?       | and have a more general set of conclusions on the response        |
| (d) Please clearly state whether all the model results of    | of TCs to future warming. We conducted several sensitivity        |
| this article are based on the output of the inner domain.    | experiments on different domain configurations and                |
|                                                              | specific experiments with adjusted southern boundaries            |
|                                                              | were also conducted (but for a different TC case that             |

|                                                                                                           |  <li>tracked further south) and it was found that the current domain configuration was optimal in terms of simulated tracks and intensity. Indicated in the manuscript (Page 6, Lines 242-244). Kindly see Supplementary Figure 1 for more details.</li> <li>(c) The northern boundary of the inner domain was also designed to consider multiple TC cases (and for further experiments, not included in this paper) that made landfall to the north of the Philippines and to appropriately simulate the subtropical ridge/Western North Pacific Sub-tropical High and Northeasterly winds. Indicated in Page 6, Lines 242-244 in the revised manuscript. Model results indicated in the manuscript are outputs of the inner domain and this</li>                                                                                                                                                                                                                                                                                                                                                                                                                                                                                                                                                                                                                                                                                                                                                                                                                                                                                                                                                                                                                                                                                                                            |
|-----------------------------------------------------------------------------------------------------------|-----------------------------------------------------------------------------------------------------------------------------------------------------------------------------------------------------------------------------------------------------------------------------------------------------------------------------------------------------------------------------------------------------------------------------------------------------------------------------------------------------------------------------------------------------------------------------------------------------------------------------------------------------------------------------------------------------------------------------------------------------------------------------------------------------------------------------------------------------------------------------------------------------------------------------------------------------------------------------------------------------------------------------------------------------------------------------------------------------------------------------------------------------------------------------------------------------------------------------------------------------------------------------------------------------------------------------------------------------------------------------------------------------------------------------------------------------------------------------------------------------------------------------------------------------------------------------------------------------------------------------------------------------------------------------------------------------------------------------------------------------------------------------------------------------------------------------------------------------------------------------------------------|
|                                                                                                           | has been indicated in the revised manuscript (Page 6, Line 238)                                                                                                                                                                                                                                                                                                                                                                                                                                                                                                                                                                                                                                                                                                                                                                                                                                                                                                                                                                                                                                                                                                                                                                                                                                                                                                                                                                                                                                                                                                                                                                                                                                                                                                                                                                                                                               |
| Lines 167-170: How do you explain your result that the simulation with the longer lead-time was the best? | Experiments with different lead times have been conducted prior
to the selection of 04 Nov 00 UTC as the initial time (longer lead-
time). Other experiments include 04 Nov 06 UTC, 12 UTC; 18
UTC; 05 Nov 00 UTC, 12 UTC; 06 Nov 00UTC, 12 UTC; and
Results of these experiments showed that this chosen initial time
with longer lead-time is able to simulate the observed track and
intensity better than later times. The longer lead-time was used
to allow for the simulation of the early stages of development of
Typhoon Haiyan, as also used by Nakamura et al. (2016) for
Typhoon Haiyan under present-day and future-climate
simulations and associated storm surge. The model initialized at
04 Nov 00UTC and 07 Nov 00UTC have simulated tracks closer
to observed (IbTRaCS). In addition, when comparing the
simulated and observed intensity (minimum sea level pressure
and maximum wind speed), it can be noted that in the time series
of the simulated intensities, the model takes longer to develop
than the observed. This is often seen in regional modeling /
limited area modeling, which seems to indicate that the model
often requires a spin-up period, for example a 36-hour spin-up
period was also implemented by Cruz and Narisma 2016 in
simulating Tropical Storm Ketsana, to reduce the effect arising
from imbalances between the simulated results due to the model
physics (microphysics, planetary boundary layer, cumulus) and
the initial and boundary conditions (Chu et al., 2018). This
behavior was found to be related to the planetary boundary and
surface layer parameterizations in WRF (Maldonado et al., 2020)
and the time needed for initialization can also be affected by the
size of the domain and terrain conditions (Chu et al., 2018). Thus, |

|                                                                                                                                                                                                                                                                                                                                                                           | we considered the time between 04 Nov 0001C to 05 Nov 12                                                                                                                                                                                                                                                                                                                                                                                                                                                                                                                                                                                                                                                                                                                                                      |
|---------------------------------------------------------------------------------------------------------------------------------------------------------------------------------------------------------------------------------------------------------------------------------------------------------------------------------------------------------------------------|---------------------------------------------------------------------------------------------------------------------------------------------------------------------------------------------------------------------------------------------------------------------------------------------------------------------------------------------------------------------------------------------------------------------------------------------------------------------------------------------------------------------------------------------------------------------------------------------------------------------------------------------------------------------------------------------------------------------------------------------------------------------------------------------------------------|
|                                                                                                                                                                                                                                                                                                                                                                           | UTC as the spin-up period (first 36 hours of simulation and at                                                                                                                                                                                                                                                                                                                                                                                                                                                                                                                                                                                                                                                                                                                                                |
|                                                                                                                                                                                                                                                                                                                                                                           | this period Haiyan was observed to be just developing from a                                                                                                                                                                                                                                                                                                                                                                                                                                                                                                                                                                                                                                                                                                                                                  |
|                                                                                                                                                                                                                                                                                                                                                                           | tropical depression to a tropical storm) and the results presented                                                                                                                                                                                                                                                                                                                                                                                                                                                                                                                                                                                                                                                                                                                                            |
|                                                                                                                                                                                                                                                                                                                                                                           | in the manuscript covers the analysis period between 5 Nov 18                                                                                                                                                                                                                                                                                                                                                                                                                                                                                                                                                                                                                                                                                                                                                 |
|                                                                                                                                                                                                                                                                                                                                                                           | UTC to 8 Nov 18 UTC to cover Haiyan's mature stage. We have                                                                                                                                                                                                                                                                                                                                                                                                                                                                                                                                                                                                                                                                                                                                                   |
|                                                                                                                                                                                                                                                                                                                                                                           | added a few lines in the revised manuscript (Page 7, Lines 253-                                                                                                                                                                                                                                                                                                                                                                                                                                                                                                                                                                                                                                                                                                                                               |
|                                                                                                                                                                                                                                                                                                                                                                           | 255). Kindly refer to Supplementary Figure 2 for more details.                                                                                                                                                                                                                                                                                                                                                                                                                                                                                                                                                                                                                                                                                                                                                |
| Line 182: Was the cumulus convection scheme                                                                                                                                                                                                                                                                                                                               | Yes, we have used the cumulus schemes in both 25km outer and                                                                                                                                                                                                                                                                                                                                                                                                                                                                                                                                                                                                                                                                                                                                                  |
| employed in both domains? Please state it clearly.                                                                                                                                                                                                                                                                                                                        | 5km inner domain. We have indicated this in the revised                                                                                                                                                                                                                                                                                                                                                                                                                                                                                                                                                                                                                                                                                                                                                       |
|                                                                                                                                                                                                                                                                                                                                                                           | manuscript (Page 7, Line 268).                                                                                                                                                                                                                                                                                                                                                                                                                                                                                                                                                                                                                                                                                                                                                                                |
| Lines 289-290 and 297-298: the mean DPE of KF                                                                                                                                                                                                                                                                                                                             | Thank you for pointing this out. The indicated DPEs in Lines                                                                                                                                                                                                                                                                                                                                                                                                                                                                                                                                                                                                                                                                                                                                                  |
| simulations is not the same in the former and latter                                                                                                                                                                                                                                                                                                                      | 289-290 (Page 11, Lines 390-391 in revised manuscript) were                                                                                                                                                                                                                                                                                                                                                                                                                                                                                                                                                                                                                                                                                                                                                   |
| lines. The same happens for the TK simulations. Please                                                                                                                                                                                                                                                                                                                    | the mean throughout the simulation period and not the analysis                                                                                                                                                                                                                                                                                                                                                                                                                                                                                                                                                                                                                                                                                                                                                |
| make the necessary corrections and update lines 562-                                                                                                                                                                                                                                                                                                                      | period. We have removed this line and retained the correct                                                                                                                                                                                                                                                                                                                                                                                                                                                                                                                                                                                                                                                                                                                                                    |
| 563 accordingly.                                                                                                                                                                                                                                                                                                                                                          | figures in Page 11, Lines 390-391 and in the conclusion (Page                                                                                                                                                                                                                                                                                                                                                                                                                                                                                                                                                                                                                                                                                                                                                 |
|                                                                                                                                                                                                                                                                                                                                                                           | 24, Lines 687-688).                                                                                                                                                                                                                                                                                                                                                                                                                                                                                                                                                                                                                                                                                                                                                                                           |
| Figure 3, x-axes: is it the simulation time or the 72-                                                                                                                                                                                                                                                                                                                    | Apologies for the confusion. We have revised the time axis of                                                                                                                                                                                                                                                                                                                                                                                                                                                                                                                                                                                                                                                                                                                                                 |
| hour verification time (as it was stated in line 171)?                                                                                                                                                                                                                                                                                                                    | Figures 3, 4 & 5 to reflect the analysis period between 18 UTC 5                                                                                                                                                                                                                                                                                                                                                                                                                                                                                                                                                                                                                                                                                                                                              |
|                                                                                                                                                                                                                                                                                                                                                                           | November 2013 to 18 UTC 8 November 2013. All experiments                                                                                                                                                                                                                                                                                                                                                                                                                                                                                                                                                                                                                                                                                                                                                      |
|                                                                                                                                                                                                                                                                                                                                                                           | were initialized at 00 UTC 4 November 2013.                                                                                                                                                                                                                                                                                                                                                                                                                                                                                                                                                                                                                                                                                                                                                                   |
| Line 319: in Figure 4 the control simulation                                                                                                                                                                                                                                                                                                                              | Thank you again for pointing this out. Same with the issue on the                                                                                                                                                                                                                                                                                                                                                                                                                                                                                                                                                                                                                                                                                                                                             |
| (KFsnOFFsf0) has a minimum mslp of about 940 hPa                                                                                                                                                                                                                                                                                                                          | DPE and we have indicated the correct figures in the revised                                                                                                                                                                                                                                                                                                                                                                                                                                                                                                                                                                                                                                                                                                                                                  |
| (not 934 hPa) and maximum wind speed less than 50                                                                                                                                                                                                                                                                                                                         | manuscript (Page 12, Lines 422-423). It now reads:                                                                                                                                                                                                                                                                                                                                                                                                                                                                                                                                                                                                                                                                                                                                                            |
| hPa (not 53.69 m/s).                                                                                                                                                                                                                                                                                                                                                      |                                                                                                                                                                                                                                                                                                                                                                                                                                                                                                                                                                                                                                                                                                                                                                                                               |
|                                                                                                                                                                                                                                                                                                                                                                           |                                                                                                                                                                                                                                                                                                                                                                                                                                                                                                                                                                                                                                                                                                                                                                                                               |
|                                                                                                                                                                                                                                                                                                                                                                           | "The control simulation (denoted as KFsnOFFsf0) has a MSLP                                                                                                                                                                                                                                                                                                                                                                                                                                                                                                                                                                                                                                                                                                                                                    |
|                                                                                                                                                                                                                                                                                                                                                                           | "The control simulation (denoted as KFsnOFFsf0) has a MSLP value of only 939 hPa and maximum wind speed of 43.47 meters                                                                                                                                                                                                                                                                                                                                                                                                                                                                                                                                                                                                                                                                                       |
|                                                                                                                                                                                                                                                                                                                                                                           | "The control simulation (denoted as KFsnOFFsf0) has a MSLP
value of only 939 hPa and maximum wind speed of 43.47 meters
per second (ms-1). Compared to the minimum central pressure                                                                                                                                                                                                                                                                                                                                                                                                                                                                                                                                                                                                                     |
|                                                                                                                                                                                                                                                                                                                                                                           | "The control simulation (denoted as KFsnOFFsf0) has a MSLP
value of only 939 hPa and maximum wind speed of 43.47 meters
per second (ms-1). Compared to the minimum central pressure
of 895hPa and 73 ms-11-min sustained wind speed in the                                                                                                                                                                                                                                                                                                                                                                                                                                                                                                                                                           |
|                                                                                                                                                                                                                                                                                                                                                                           | "The control simulation (denoted as KFsnOFFsf0) has a MSLP
value of only 939 hPa and maximum wind speed of 43.47 meters
per second (ms-1). Compared to the minimum central pressure
of 895hPa and 73 ms-11-min sustained wind speed in the
observations, this is a difference of 38 hPa and 29.53ms-1,                                                                                                                                                                                                                                                                                                                                                                                                                                                                                            |
|                                                                                                                                                                                                                                                                                                                                                                           | "The control simulation (denoted as KFsnOFFsf0) has a MSLP
value of only 939 hPa and maximum wind speed of 43.47 meters
per second (ms-1). Compared to the minimum central pressure
of 895hPa and 73 ms-11-min sustained wind speed in the
observations, this is a difference of 38 hPa and 29.53ms-1,
respectively."                                                                                                                                                                                                                                                                                                                                                                                                                                                                          |
| Figure 4: For consistency with the symbols of the other                                                                                                                                                                                                                                                                                                                   | "The control simulation (denoted as KFsnOFFsf0) has a MSLP
value of only 939 hPa and maximum wind speed of 43.47 meters
per second (ms-1). Compared to the minimum central pressure
of 895hPa and 73 ms-11-min sustained wind speed in the
observations, this is a difference of 38 hPa and 29.53ms-1,
respectively."
As suggested, we have revised the figures for better                                                                                                                                                                                                                                                                                                                                                                                                                  |
| Figure 4: For consistency with the symbols of the other experiments, it is suggested to change the pattern of                                                                                                                                                                                                                                                             | "The control simulation (denoted as KFsnOFFsf0) has a MSLP
value of only 939 hPa and maximum wind speed of 43.47 meters
per second (ms-1). Compared to the minimum central pressure
of 895hPa and 73 ms-11-min sustained wind speed in the
observations, this is a difference of 38 hPa and 29.53ms-1,
respectively."
As suggested, we have revised the figures for better
representation of the different experiments. Kindly refer to the                                                                                                                                                                                                                                                                                                                                              |
| Figure 4: For consistency with the symbols of the other
experiments, it is suggested to change the pattern of
TKsnOFFsf1 to dotted line. In the current figure it is                                                                                                                                                                                                | "The control simulation (denoted as KFsnOFFsf0) has a MSLP
value of only 939 hPa and maximum wind speed of 43.47 meters
per second (ms-1). Compared to the minimum central pressure
of 895hPa and 73 ms-11-min sustained wind speed in the
observations, this is a difference of 38 hPa and 29.53ms-1,
respectively."
As suggested, we have revised the figures for better
representation of the different experiments. Kindly refer to the
updated Figure 4 (Page 13) in the revised manuscript.                                                                                                                                                                                                                                                                                     |
| Figure 4: For consistency with the symbols of the other
experiments, it is suggested to change the pattern of
TKsnOFFsf1 to dotted line. In the current figure it is
difficult to distinguish it from TKsnOFFsf0.                                                                                                                                                | "The control simulation (denoted as KFsnOFFsf0) has a MSLP
value of only 939 hPa and maximum wind speed of 43.47 meters
per second (ms-1). Compared to the minimum central pressure
of 895hPa and 73 ms-11-min sustained wind speed in the
observations, this is a difference of 38 hPa and 29.53ms-1,
respectively."
As suggested, we have revised the figures for better
representation of the different experiments. Kindly refer to the
updated Figure 4 (Page 13) in the revised manuscript.                                                                                                                                                                                                                                                                                     |
| Figure 4: For consistency with the symbols of the other
experiments, it is suggested to change the pattern of
TKsnOFFsf1 to dotted line. In the current figure it is
difficult to distinguish it from TKsnOFFsf0.
Lines 349-350: in figure 6 the RMSE of KFsnOFFsf1                                                                                           |  <li>"The control simulation (denoted as KFsnOFFsf0) has a MSLP value of only 939 hPa and maximum wind speed of 43.47 meters per second (ms-1). Compared to the minimum central pressure of 895hPa and 73 ms-11-min sustained wind speed in the observations, this is a difference of 38 hPa and 29.53ms-1, respectively."</li> <li>As suggested, we have revised the figures for better representation of the different experiments. Kindly refer to the updated Figure 4 (Page 13) in the revised manuscript.</li>                                                                                                                                                                                                                                                                                 |
| Figure 4: For consistency with the symbols of the other
experiments, it is suggested to change the pattern of
TKsnOFFsf1 to dotted line. In the current figure it is
difficult to distinguish it from TKsnOFFsf0.
Lines 349-350: in figure 6 the RMSE of KFsnOFFsf1
is about 10 m/s and its correlation is between 0.8 and                                 |  <li>"The control simulation (denoted as KFsnOFFsf0) has a MSLP value of only 939 hPa and maximum wind speed of 43.47 meters per second (ms-1). Compared to the minimum central pressure of 895hPa and 73 ms-11-min sustained wind speed in the observations, this is a difference of 38 hPa and 29.53ms-1, respectively."</li> <li>As suggested, we have revised the figures for better representation of the different experiments. Kindly refer to the updated Figure 4 (Page 13) in the revised manuscript.</li> <li>Thank you for spotting this. We have indicated the correct figures in the revised manuscript (Page 15, Lines 458-459). It</li>                                                                                                                                              |
| Figure 4: For consistency with the symbols of the other
experiments, it is suggested to change the pattern of
TKsnOFFsf1 to dotted line. In the current figure it is
difficult to distinguish it from TKsnOFFsf0.
Lines 349-350: in figure 6 the RMSE of KFsnOFFsf1
is about 10 m/s and its correlation is between 0.8 and
0.85 (i.e. lower than 0.89). | "The control simulation (denoted as KFsnOFFsf0) has a MSLP
value of only 939 hPa and maximum wind speed of 43.47 meters
per second (ms-1). Compared to the minimum central pressure
of 895hPa and 73 ms-11-min sustained wind speed in the
observations, this is a difference of 38 hPa and 29.53ms-1,
respectively."
As suggested, we have revised the figures for better
representation of the different experiments. Kindly refer to the
updated Figure 4 (Page 13) in the revised manuscript.
Thank you for spotting this. We have indicated the correct
figures in the revised manuscript (Page 15, Lines 458-459). It
now reads:                                                                                                                                       |
| Figure 4: For consistency with the symbols of the other
experiments, it is suggested to change the pattern of
TKsnOFFsf1 to dotted line. In the current figure it is
difficult to distinguish it from TKsnOFFsf0.
Lines 349-350: in figure 6 the RMSE of KFsnOFFsf1
is about 10 m/s and its correlation is between 0.8 and
0.85 (i.e. lower than 0.89). |  <li>"The control simulation (denoted as KFsnOFFsf0) has a MSLP value of only 939 hPa and maximum wind speed of 43.47 meters per second (ms-1). Compared to the minimum central pressure of 895hPa and 73 ms-11-min sustained wind speed in the observations, this is a difference of 38 hPa and 29.53ms-1, respectively."</li> <li>As suggested, we have revised the figures for better representation of the different experiments. Kindly refer to the updated Figure 4 (Page 13) in the revised manuscript.</li> <li>Thank you for spotting this. We have indicated the correct figures in the revised manuscript (Page 15, Lines 458-459). It now reads:</li>                                                                                                                                   |
| Figure 4: For consistency with the symbols of the other
experiments, it is suggested to change the pattern of
TKsnOFFsf1 to dotted line. In the current figure it is
difficult to distinguish it from TKsnOFFsf0.
Lines 349-350: in figure 6 the RMSE of KFsnOFFsf1
is about 10 m/s and its correlation is between 0.8 and
0.85 (i.e. lower than 0.89). |  <li>"The control simulation (denoted as KFsnOFFsf0) has a MSLP value of only 939 hPa and maximum wind speed of 43.47 meters per second (ms-1). Compared to the minimum central pressure of 895hPa and 73 ms-11-min sustained wind speed in the observations, this is a difference of 38 hPa and 29.53ms-1, respectively."</li> <li>As suggested, we have revised the figures for better representation of the different experiments. Kindly refer to the updated Figure 4 (Page 13) in the revised manuscript.</li> <li>Thank you for spotting this. We have indicated the correct figures in the revised manuscript (Page 15, Lines 458-459). It now reads:</li> <li>"Of all the simulations, the simulation with the combination of</li>                                                          |
| Figure 4: For consistency with the symbols of the other
experiments, it is suggested to change the pattern of
TKsnOFFsf1 to dotted line. In the current figure it is
difficult to distinguish it from TKsnOFFsf0.
Lines 349-350: in figure 6 the RMSE of KFsnOFFsf1
is about 10 m/s and its correlation is between 0.8 and
0.85 (i.e. lower than 0.89). |  <li>"The control simulation (denoted as KFsnOFFsf0) has a MSLP value of only 939 hPa and maximum wind speed of 43.47 meters per second (ms-1). Compared to the minimum central pressure of 895hPa and 73 ms-11-min sustained wind speed in the observations, this is a difference of 38 hPa and 29.53ms-1, respectively."</li> <li>As suggested, we have revised the figures for better representation of the different experiments. Kindly refer to the updated Figure 4 (Page 13) in the revised manuscript.</li> <li>Thank you for spotting this. We have indicated the correct figures in the revised manuscript (Page 15, Lines 458-459). It now reads:</li> <li>"Of all the simulations, the simulation with the combination of KF and sf1 without nudging have the lowest RMSE (22 hPa</li>  |

|                                                          | coefficient of 0.78 and 0.82 for MSLP and maximum winds,                            |
|----------------------------------------------------------|-------------------------------------------------------------------------------------|
|                                                          | respectively."                                                                      |
| Lines 351-352: in figure 6 the RMSE of TKsnONsf0 is      | Thank you for spotting this. We have indicated the correct                          |
| about 15 m/s and its correlation is about 0.69.          | figures in the revised manuscript (Page 15, Line 459-461). It now                   |
|                                                          | reads:                                                                              |
|                                                          |                                                                                     |
|                                                          | "While the simulation with the poorest performance i.e. highest              |
|                                                          | RMSE (37 hPa and 14.17 ms -1 ) and lowest correlation coefficient |
|                                                          | (0.60 and 0.69 for MSLP and maximum winds, respectively) is                         |
|                                                          | the simulation with the combination of TK, sf0, with spectral                       |
|                                                          | nudging turned on."                                                                 |
| Line 409: The simulation with the closest landfall time  | Thank you for spotting this. We have indicated the correct                          |
| is not shown in Table 3, but it can be derived by Figure | figures in the revised manuscript (Page 17, Line 518).                              |
| 11 (as far as the experiments without spectral nudging   |                                                                                     |
| are concerned).                                          |                                                                                     |
| Line 464: Please justify your choice to present only the | For improved readability, we have chosen to present the                             |
| runs without nudging in figure 11.                       | experiments without nudging to represent the TC-associated                          |
|                                                          | rainfall in the different experiments. Similar rainfall patterns                    |
|                                                          | were found in the experiments with nudging as shown in                              |
|                                                          | Supplementary Figure 4.                                                             |
| Line 488: the steering flow bias has not been shown in   | Thank you for pointing this out. We have removed this line in                       |
| figure 12.                                               | the manuscript (Page 21, Line 600).                                                 |
| Figures 12 and 14: Did you interpolate the WRF output    | Yes, the 6-hourly WRF output was interpolated to the coarser                        |
| to the coarser ERA5 grid? Which interpolation method     | ERA5 grid using First-order Conservative Remapping through                          |
| did you use? Please include this information in the      | CDO's remapcon function. We have specified this in the revised                      |
| article.                                                 | manuscript (Page 21, Lines 611-613; Page 22, 645-647; Page 23,                      |
|                                                          | Lines 665-667)                                                                      |
| Figures 12, 13, 14: (a) Please justify the use of the    | (a) KFsnOFFsf1 and TKsnOFFsf1 were used in this section to                          |
| KFsnOFFsf1 and TKsnOFFsf1 experiments instead of         | represent the experiments with KF and TK runs, primarily to                         |
| all the KF and TK runs. (b) are these figures based on   | save on space but more importantly, similar results were found                      |
| 6-hourly ERA5 and WRF output?                            | in the average of the experiments using KF and TK as cumulus                        |
|                                                          | convection scheme. Kindly refer to Supplementary Figures 5,6                        |
|                                                          | and 7.                                                                              |
|                                                          | (b) Yes, these are based on 6-hourly ERA5 and WRF output                            |
|                                                          | values. We have indicated this in the revised manuscript (Page                      |
|                                                          | 21, Lines 611-613; Page 22, 645-647; Page 23, Lines 665-667)                        |
| Lines 501-502: Please clarify in the article whether the | The vertical wind shear was re-computed by averaging the                            |
| vertical wind shear was computed (a) from time-          | instantaneous values of the vertical wind shear (i.e. firstly                       |
| averaged u and v winds at 200 and 850 hPa (i.e. firstly  | calculating the instantaneous vertical wind shear at each grid-                     |
| calculating the time-averaged u and v at each grid-      | point and then calculating its time-average value) (kindly see                      |
| point and then using them to calculate the vertical wind | revised Figure 12). We specified this in the revised manuscript                     |
| shear), or (b) by averaging the instantaneous values of  | (Page 22, Lines 619-620)                                                            |
| the vertical wind shear (i.e. firstly calculating the    |                                                                                     |

| instantaneous vertical wind shear at each grid-point     |                                                                |
|----------------------------------------------------------|----------------------------------------------------------------|
| and then calculating its time-average value).            |                                                                |
| Line 536: (a) do you mean that KF shows a higher         | (a) Thank you for pointing this out. We have corrected this    |
| relative humidity along the track? Otherwise, it         | observation in the manuscript (Page 23, Lines 657-658).        |
| disagrees with the previous discussion in this           | (b) We think this is a great suggestion so we have revised the |
| paragraph. (b) for clarity it is suggested to draw the   | figure to show the simulated tracks. Please refer to the new   |
| tracks of the simulated and actual tracks on both panels | Figure 12 in the revised manuscript.                           |
| of figure 14.                                            |                                                                |
| Technical corrections:                                   |                                                                |
| Line 152: " and model physics (Isaksen et al.,           | Revised in the manuscript (Page 6, Line 230)                   |
| 2010)."                                                  |                                                                |
| Line 158: " different parameterization"                  | Revised in the manuscript (Page 6, Line 239)                   |
| Line 165: It is a 180-hour period (not 174-hour) from    | Revised in the manuscript (Page 6, Line 248)                   |
| 00 UTC 4 November to 12 UTC 11 November.                 |                                                                |
| Line 175: " is bounded by 100-170 degrees East"          | Revised in the manuscript (Page 7, Line 260)                   |
| Line 251: " maximum 10m winds to evaluate"               | Revised in the manuscript (Page 10, Line 352)                  |
| Line 268: " relative vorticity maxima"                   | Revised in the manuscript (Page 10, Lines 368-369)             |
| Line 286: " without nudging (snOFF)"                     | Revised in the manuscript (Page 11, Line 387)                  |
| Line 312: " of the DPE (km)"                             | Revised in the manuscript (Page 12, Line 414)                  |
| Line 473: " the KF scheme shows"                         | Revised in the manuscript (Page 20, Line 582)                  |
| Lines 496, 527, 543: KFsnOFFsd1 and TKsnOFFsd1           | Revised in the manuscript (Lines 610, 645, 665)                |
| must be corrected to KFsnOFFsf1 and TKsnOFFsf1,          |                                                                |
| respectively.                                            |                                                                |

**References**

Chu, Q., Xu, Z., Chen, Y. and Han, D.W.: Evaluation of the ability of the Weather Research and Forecasting model to reproduce a sub-daily extreme rainfall event in Beijing, China using different domain configurations and spin-up times. Hydrology and Earth System Sciences Discussions, 22, 3391–3407. https://doi.org/10.5194/hess-22-3391-2018, 2018.

Cruz, F. and Narisma, G.: WRF simulation of the heavy rainfall over Metropolitan Manila, Philippines during tropical cyclone Ketsana: a sensitivity study. Meteorology and Atmospheric Physics, 128(4), 415–428. https://doi:10.1007/s00703-015-0425-x, 2016.

Maldonado T, Amador JA, Rivera ER, Hidalgo HG, Alfaro EJ.: Examination of WRF-ARW Experiments Using Different Planetary Boundary Layer Parameterizations to Study the Rapid Intensification and Trajectory of Hurricane Otto (2016). Atmosphere, 11(12):1317. https://doi.org/10.3390/atmos11121317, 2020.

Nakamura, R., Shibayama, T., Esteban, M. and Iwamoto, T.: Future typhoon and storm surges under different global warming scenarios: case study of typhoon Haiyan (2013). Nat. Hazards, 82, 1645-1681. https://doi:10.1007/s11069-016-2259-3, 2016.

**Recommendation: MAJOR REVISION**

| Reviewer 2's COMMENTS                                 | Authors' RESPONSES                                              |
|--------------------------------------------------------------|-----------------------------------------------------------------|
| General comments                                             |                                                                 |
| The authors utilized WRF-ARW to simulate Typhoon             | Dear Referee,                                                   |
| Haiyan and investigate the role of cumulus convection        |                                                                 |
| (KF and TK schemes), surface flux parameterizations,         | Thank you very much for highlighting the importance of our      |
| spectral nudging, and initial and boundary conditions        | work, the useful feedback on the submitted manuscript, and for  |
| (ERA5 and EDA). They concluded that the TK scheme            | giving us the opportunity to submit a much improved version     |
| and spectral nudging improve track simulations with          | of the manuscript. We have addressed the major and minor        |
| lower mean DPE than the other model configurations.          | concerns raised. All changes are highlighted in the revised     |
| On the other hand, KF scheme and varying the surface         | manuscript and line numbers refer to the revised manuscript     |
| flux options improve the intensity.                          | with tracked changes.                                           |
| This type of study will definitely be of a great addition to |                                                                 |
| works that optimize a model's configuration of TC            | Please see below our specific responses and refer to the        |
| simulations in the Philippines, but in its current form is   | attached revised manuscript and supplementary file for more     |
| not yet ready for publication. Major parts of the paper      | details.                                                        |
| should be rewritten due to the following major concerns:     |                                                                 |
| Major Concerns:                                              |                                                                 |
| 1. (Line 55~Line 105, Line 125) Although a future            | Thank you for pointing this out. The overall approach of the    |
| plan for conducting pseudo-global warming simulations        | study is that we have used WRF configured as NWP to get the     |
| was mentioned, WRF-ARW was used in the paper as a            | best configuration for hindcast TC case simulations and         |
| numerical weather prediction (NWP) model to simulate a       | eventually use that configuration to simulate the TC cases with |
| weather event (TC Haiyan). However, the literature           | future climate forcings. The results included in this paper are |
| review (introduction) seems to interchange regional          | from the former i.e. as a sensitivity study using Typhoon       |
| climate models (climatological simulations) with             | Haiyan as the TC case. We have revised the manuscript to        |
| numerical weather prediction models (short-term              | make the distinction clearer i.e. studies with NWP event-based  |
| weather events) resulting in mixed and improper              | hindcast simulations to build a foundation on sensitivities to  |
| citations of papers that use RCMs and NWPs. Event            | model parameterizations and settings. We have also cited some   |
| simulations are different from climatological runs.          | studies using WRF as LAM with future climate forcings as        |
| Although WRF and other NWPs can also be used as              | initial and lateral boundary conditions in support of the       |
| RCM, they are usually modified to efficiently work for       | rationale behind the bigger study. Significant revisions were   |
| climatological simulations (e.g. CLWRF, RegCM                | made in Pages 2-4, Lines $55 - 170$ in the revised manuscript.  |
| RCM version of MM5, NHRCM – RCM version of                   |                                                                 |
| JMA/MRI NHM). NHRCM, and not WRF, is the model               | Apologies for this mistake. Cruz et al 2016 should read Cruz    |
| used by Cruz et al., 2016 in Line 132.                       | and Narisma 2016. We have revised this in Page 1, Lines 71-     |
|                                                              | 72 and included in references of the revised manuscript.        |
| The paper literature review should focus on studies that     | We have included additional discussion in the introduction,     |
| conduct TC short-term simulations using models (e.g.         | particularly that of surface flux options e.g. from a study by  |
| WRF, NHM) that are considered as NWP and not RCM.            | Kueh et al., 2019 using WRF (Page 3, Lines 88-102 of the        |
|                                                              | revised manuscript). Additional studies on ICBC (Islam et al.,  |
|                                                              | 2015; Mohanty et al., 2010; Shepherd and Walsh, 2016) and       |

| The literature review also fell short in terms of            | spectral nudging (in WRF as NWP Mori et al., 2014; Kueh et          |
|--------------------------------------------------------------|---------------------------------------------------------------------|
| discussing studies that tackle the other sensitivity         | al., 2019 and as RCM Shen et al., 2017; Cha et al., 2011) have      |
| parameters such as spectral nudging, surface flux, and       | also been added in the introduction section (Pages 2-3, Lines       |
| ICBC. The reviewer hopes to see a clearer revised            | 88-123).                                                            |
| Introduction with an additional review on the said           |                                                                     |
| parameters.                                                  |                                                                     |
| 2. The objective and analysis of this paper are very  | Thank you for these clarifications. There is only one               |
| promising but the initial forcing is also very critical to   | mother/outer domain (D01) and child/inner (D02) domain and          |
| consider it as a sensitivity analysis. Kindly clarify if the | the same domain settings were used in all the sensitivity           |
| researchers downscaled only one mother domain (D1)           | experiments (as shown in Figure 1 of the submitted                  |
| for all D2 sensitivity runs? If not, then it will be         | manuscript). The same physics parameterizations were also           |
| inappropriate and difficult to compare the sensitivity of    | used in both outer (D01) and inner (D02) domains. We have           |
| TC track and intensity to parameterizations if the initial   | explicitly indicated these in the text (Page 6, Lines 235-245)      |
| forcing (D1) for each experiment have different model        | and in Table 3 of the revised manuscript.                           |
| physics. This might explain the different (or larger         |                                                                     |
| differences of) values of intensities at t=0 in Figure 4.    | Since we are using two-way nesting and there is feedback from       |
| The reviewer strongly suggests to reconsider rerunning       | the outer to the inner domain and vice versa, it is important that  |
| all simulations using only one D1 simulation as forcing      | the same physics parameterization is used in both domains.          |
| to all D2 experiments.                                       | This is the used in WRF with multiple and nested domains            |
|                                                              | (Werner and Wang, 2017; Dudhia 2015), as there could be             |
|                                                              | issues with two-way nesting when physics parameterization           |
|                                                              | differs across the nest boundaries (e.g. in precipitation fields of |
|                                                              | the mother/outer domain) (Dudhia 2015) and used in past             |
|                                                              | studies (e.g. Wang and Wang, 2014; Islam et al., 2015). The         |
|                                                              | physics parameterization, particularly the cumulus scheme,          |
|                                                              | was changed in each sensitivity experiment in both domains.         |
|                                                              |                                                                     |
|                                                              | Apologies for the confusion. We have revised the time axis of       |
|                                                              | Figures 3, 4 & 5 to reflect the analysis period between 18 UTC      |
|                                                              | 5 November 2013 to 18 UTC 8 November 2013. All                      |
|                                                              | experiments were initialized at 00 UTC 4 November 2013              |
|                                                              | (t=0). The different values of intensities at the start of the      |
|                                                              | analysis period (18 UTC 5 November 2013) is expected since          |
|                                                              | there has already been interaction between D01 & D02. The           |
|                                                              | same initial conditions were used for D01 and D02. There is no      |
|                                                              | difference in the simulated intensity (MSLP = 1005hPa; max          |
|                                                              | winds = 17 m/s) at t=0 (04 Nov 00 UTC) for both mother/outer        |
|                                                              | domain (D01) and child/inner domain (D02) for all sensitivity       |
|                                                              | experiment. Kindly refer to Supplementary Figure 3 for more         |
|                                                              | details.                                                            |
|                                                              |                                                                     |
|                                                              | Given this clarification, there is no need to rerun the             |
|                                                              | simulations.                                                        |

| With this 2nd major concern, it will be difficult to give | Given what we have explained above, there is no reason for the   |
|-----------------------------------------------------------|------------------------------------------------------------------|
| meaningful comments on the results and discussions.       | 2 nd concern.                                         |
| 3. (Line 155-163, 166). Kindly provide supplementary      | Thank you for this suggestion. We have included some figures     |
| materials for the results of the other domain             | in the supplementary material. Initial simulations have been     |
| configurations that led the authors to select the control | done to check model performance using different domain           |
| run model setup. These supplementary materials are very   | configurations and horizontal resolution i.e. (a) single domain  |
| important to justify the model setup of the control run.  | (at 12km horizontal resolution); (b) two domains (at 12 and      |
|                                                           | 4km horizontal resolution); (c) same as (b) but with bigger      |
|                                                           | inner domain; (d) three domains (12, 4 and 1.3km horizontal      |
|                                                           | resolution); and (e) two domains (25, 5km) horizontal            |
|                                                           | resolution. Domain configuration (e) was used for the            |
|                                                           | sensitivity experiments which simulated the lowest minimum       |
|                                                           | sea level pressure and maximum winds, and in consideration       |
|                                                           | of computing resources and other TC cases that were simulated    |
|                                                           | in the project. Kindly refer to Supplementary Figure 1 for more  |
|                                                           | details.                                                         |
|                                                           |                                                                  |
|                                                           | Experiments with different lead times have also been             |
|                                                           | conducted prior to the selection of 04 Nov 00 UTC as the initial |
|                                                           | time (longer lead-time) as well as experiments on different      |
|                                                           | domain configurations and specific experiments with adjusted     |
|                                                           | southern boundaries were also conducted (but for a different     |
|                                                           | TC case that tracked further south) Kindly refer to              |
|                                                           | Supplementary Figures 1b and 2 for more details.                 |
|                                                           |                                                                  |
|                                                           | For the choice of cumulus parameterization in the control run,   |
|                                                           | we have chosen KF for the control run since it's used by         |
|                                                           | PAGASA in its NWP configuration; the default surface flux        |
|                                                           | option (isftcflx = 0) and no spectral nudging so that we can     |
|                                                           | easily assess the sensitivity to these physics parameterization  |
|                                                           | and alternative model options. Other parameterizations were      |
|                                                           | based on previous work on Typhoon Haiyan i.e. Li et al., 2018.   |
| Minor suggestions                                         |                                                                  |
| (Line 113): Correct the year "2012" to "2013".            | Thank you for spotting this. Revised in Page 5, Line 190 in the  |
|                                                           | revised manuscript                                               |
| (Line 125): Kindly reconsider "NWP" instead of            | The overall approach of the study is that we have used WRF       |
| "RCM".                                                    | configured as NWP to get the best configuration for hindcast     |
|                                                           | TC cases simulations and eventually use that configuration to    |
|                                                           | simulate the TC cases with future climate forcings. The results  |
|                                                           | included in this paper is from the former i.e. as a sensitivity  |
|                                                           | study using Typhoon Haiyan as the TC case. We have revised       |
|                                                           | the manuscript to indicate that we used WRF as a LAM so as       |
|                                                           | to avoid confusion.                                              |

| There is no "Powers 2016" in the references.              | Apologies for this. Powers 2016 should read Powers 2017. We        |
|-----------------------------------------------------------|--------------------------------------------------------------------|
|                                                           | have revised in Page 5, Line 204 in the revised manuscript and     |
|                                                           | already indicated in the references.                               |
| (Line 132): Cruz et al., 2016 uses NHRCM and not          | Apologies for this mistake. Cruz et al 2016 should read Cruz       |
| WRF to make temperature and rainfall projections in the   | and Narisma 2016. We have revised this in Page 4, Line 210         |
| Philippines.                                              | and included in references of the revised manuscript.              |
| (Line 155-170): Kindly provide a table for your control   | We have used two-way nesting (between the outer domain D01         |
| run's model setup as indicated in this section. Make sure | and inner domain D02) with horizontal resolution of 25km for       |
| to clarify if you performed one-way or two-way nesting,   | D01 and 5km for D01; and 44 vertical levels with model top of      |
| specify the input forcing, temporal and spatial           | 50hPa. We have explicitly indicated this in the manuscript and     |
| resolutions (dt,dx,dy,dz), model physics, and so on.      | added a table for easier reference. Please refer to Table 3, Pages |
|                                                           | 9-10 of the revised manuscript.                                    |
| (Line 180): "These cumulus schemes are used because       | PAGASA uses KF, and TK is used for tropical ocean                  |
| PAGASA uses KF". Does PAGASA also uses TK?                | applications. We have indicated this in Page 7, Lines 266-273      |
| Does the writer mean "The KF cumulus scheme was           | in the revised manuscript.                                         |
| used because"?                                            |                                                                    |
| (Line 185): There is no Sun et al., 2019 in the           | Thank you for spotting this. Should read and have added Sun        |
| references.                                               | et al., 2015 in the text and references.                           |
| The discussion on TK is too short and vague. The author   | We have revised and added in the discussion on cumulus             |
| should also provide short discussion of the main output   | parameterization particularly on Tiedtke scheme, and added a       |
| of the cited references. Same comment for Lines 194-      | brief description on the outputs of the cited references. Kindly   |
| 195, 205.                                                 | see Page 7, Lines 274-284.                                         |
| (Line 206): Check repeating phrases in the sentence with  | Thank you for pointing this out. We have revised this in Page      |
| "Charnock's (1995)".                                      | 8, Line 303.                                                       |

**References**

Cha, D-H., Jin, C-S., Lee, D-K., Kuo, Y-H.: Impact of intermittent spectral nudging on regional climate simulation using Weather Research and Forecasting model. Journal of Geophysical Research: Atmospheres. https://https://doi.org/10.1029/2010JD015069, 2011.

Cruz, F. and Narisma, G.: WRF simulation of the heavy rainfall over Metropolitan Manila, Philippines during tropical cyclone Ketsana: a sensitivity study. Meteorology and Atmospheric Physics, 128(4), 415–428. https://doi:10.1007/s00703-015-0425-x, 2016.

Dudhia,J.:OverviewofWRFPhysics.Retrievedfromhttp://homepages.see.leeds.ac.uk/~lecag/wiser/sample\_wiser\_files.dir/Physics\_Dudhia.ppt.pdf2015.

Li, F., Song, J. and Li, X.: A preliminary evaluation of the necessity of using a cumulus parameterization scheme in high-resolution simulations of Typhoon Haiyan (2013). Nat Hazards 92, 647–67. https://doi:10.1007/s11069-018-3218-y, 2018.

Kueh, M.-T., Chen, W.-M., Sheng, Y.-F., Lin, S. C., Wu, T.-R., Yen, E., Tsai, Y.-L., and Lin, C.-Y.: Effects of horizontal resolution and air–sea flux parameterization on the intensity and structure of simulated Typhoon Haiyan (2013), Nat. Hazards Earth Syst. Sci., 19, 1509–1539, https://doi:10.5194/nhess-19-1509-2019, 2019.

Islam, T., Srivastava, P.K., Rico-Ramirez, M.A. et al.: Tracking a tropical cyclone through WRF–ARW simulation and sensitivity of model physics. Nat Hazards 76, 1473–1495, https://doi:10.1007/s11069-014-1494-8, 2015.

Mohanty, U. C., Osuri, K. K., Routray, A., Mohapatra, M., and Pattanayak, S.: Simulation of bay of bengal tropical cyclones with wrf model: Impact of initial and boundary conditions. Marine Geodesy, 33(4), 294–314. https://doi:10.1080/01490419.2010.518061, 2010. Mori, N., Kato, M., Kim, S., Mase, H., Shibutani, Y., Takemi, T., Tsuboki, K., and Yasuda, T.: Local amplification of storm surge by Super Typhoon Haiyan in Leyte Gulf. Geophysical Research Letters, Volume 41, Issue 14, Pages 5106-5113. https://doi:10.1002/2014GL060689, 2014.

Shen, W., Tang, J., Wang, Y., Wang, S., and Niu, X.: Evaluation of WRF model simulations of tropical cyclones in the western North Pacific over the CORDEX East Asia domain. Climate Dynamics, 48(7), 2419–2435. https://doi:10.1007/s00382-016-3213-5, 2017.

Shepherd, T. J., and Walsh, K. J.: Sensitivity of hurricane track to cumulus parameterization schemes in the WRF model for three intense tropical cyclones: impact of convective asymmetry. Meteorog. Atmos. Phys., 129(4), 345–374. https://doi:10.1007/s00703-016-0472-y, 2017.

Werner, K. and Wang, G: Nesting in WRF. Climate Science Org Australia. Retrieved from /https://www.climatescience.org.au/sites/default/files/werner\_nesting.pdf, 2017.

**Supplement of the response to comments of Reviewers on**

**Sensitivity of simulating Typhoon Haiyan (2013) using WRF: the role of cumulus convection, surface flux parameterizations, spectral nudging, and initial and boundary conditions**

Delfino et. al.

This supplement contains figures to support responses to comments of Anonymous Referee #1&2.

The overall approach of this study is to have a common domain for multiple TC cases in this region (other TC cases not included in this paper, but are the focus of a follow-on paper, about to be submitted) to understand and have a more general set of conclusions on the response of TCs to future warming. Initial simulations have been done to check model performance using different domain configurations and horizontal resolution i.e. (a) single domain (at 12km horizontal resolution); (b) two domains (at 12 and 4km horizontal resolution); (c) same as (b) but with bigger inner domain; (d) three domains (12, 4 and 1.3km horizontal resolution); and (e) two domains (25,5km) horizontal resolution. Domain configuration (e) was used for the sensitivity experiments which simulated the lowest minimum sea level pressure and maximum winds, and in consideration of computing resources and other TC cases that were simulated in the project.

---

## Author Response (AR2)

*Response to comments of **Anonymous Referee 1 & 2** on*

**Sensitivity of simulating Typhoon Haiyan (2013) using WRF: the role of cumulus convection, surface flux parameterizations, spectral nudging, and initial and boundary conditions**

Delfino et. al.

**Anonymous Referee #1 [**Report #2 Submitted on 19 Jul 2022**]**

| Editor's COMMENTS | Authors' RESPONSES |
|---|---|
| Dear authors
although the manuscript has improved a lot with the provided revisions there are: (a) some technical corrections which should be made and b) most importantly a concern on the use of 2-way nesting in the sensitivity experiments. For that reason i would like to invite you to revise your manuscripts addressing the reviewers comments. Your manuscript will be further reviewed by the editor and the referees. We thank you for your efforts that aim at improving the manuscript and bring it to the NHESS standards. | Dear Editor,
Thank you very much for giving us the opportunity to submit an improved version of the manuscript. We appreciate the thoroughness and objectiveness of the comments and have addressed the specific concerns raised by the Reviewers, particularly the concern on the use of two-way nesting. Please note that, consistent with our original plan for this study, we are not going to attempt to build a second study based on the use of 1-way nesting, in the study of tropical cyclones: we have provided ample literature review to support our decision, as well as the expert opinion of WRF developers. Please refer to our response to Reviewer 2 for more details.

All changes are highlighted in the revised manuscript. All line numbers refer to the revised manuscript with tracked changes. |
| **Reviewer 1's COMMENTS** | **Authors' RESPONSES** |
| **General comments** | |
| **accepted subject to technical corrections.** | Thank you very much for the positive review. We have made the suggested technical corrections in the revised manuscript. Please see below our specific responses and refer to the attached revised manuscript for more details. |
| The authors have addressed successfully the suggested corrections. The article will be acceptable for publication after a few technical corrections are performed. | |
| **Suggested corrections:** | |
| Line 130: please insert a bullet at the beginning of the question (similarly to the question above). | Revised in Lines 128 – 131 of the revised manuscript |
| Line 206: "… and 5-30 degrees North.". | Revised in Line 238 of the revised manuscript |

| | |
|---|---|
| Table 1, 1st row (2nd column) and 2nd line of the caption of Figure 4: "Fritsch". | Revised in Table 1, 1st Row/2nd Column, of the revised manuscript |
| Line 281: "… domains, include:". | Revised in Line 322 of the revised manuscript |
| Line 283 and 10th row (right column) of Table 3: "Obukhov". | Revised in Table 3 of the revised manuscript |
| Table 3, 10th row (left column): "Surface Layer" instead of "180-hour period". | Revised in Table 3 of the revised manuscript |
| Line 342 and lines 351-352: It is mentioned in line 342 that the simulations using the TK scheme have a mean DPE of 47 km. This mean value is calculated from the average of the 3 TK simulations with nudging (TKsnONsf0, TKsnONsf1, TKsnONsf2) and the 3 TK simulations without nudging (TKsnOFFsf0 TKsnOFFsf1 TKsnOFFsf2). However, lines 351-352 mention that the mean DPE of the former TK simulations (with nudging) is 68 km while the mean DPE of the latter simulations (without nudging) is 87km. Both errors (68 km, 87 km) are above the overall TK mean of 47 km. Which errors are not correct? | Thank you very much for spotting this. We have corrected the figures in Lines 393 of the revised manuscript. |
| Lines 359-360: The sentence "The one and two black stars … land, respectively" must be removed because the single and double stars do not exist in this figure in the revised version of the article. | We have removed these in Lines 400-401 of the revised manuscript. |
| Line 366: In Figure 4b the maximum wind speed of the control simulation KFsnOFFsf0 is stronger than 45 m/s (and less than 50 m/s). After this correction, please also correct its difference from the observed max wind speed in line 367. | Thank you very much for spotting this. Revised in Lines 407-408 of the revised manuscript. |
| Line 367: There is a difference of 44 hPa (and not 38 hPa) between the min mslp of KFsnOFFsf0 (939 hPa) and the observed min mslp (895 hPa) of the typhoon. | Thank you very much for spotting this. Revised in Line 408 of the revised manuscript. |
| Figure 4: The lower row of panels is identical to the upper one. | Apologies for this, the lower and upper panels are merely duplicates. We have removed the lower panel in the revised manuscript. |
| Line 399: "… without nudging has …". | Revised in Line 441 of the revised manuscript |
| Line 517: "… KF scheme (Fig. 11 b-d) than those that used TK scheme (Fig. 11 e-g).". | Revised in Line 560 of the revised manuscript |

**Anonymous Referee #2** [Report #1, Submitted on 6 July 2022]

| Reviewer 2's COMMENTS | Authors' RESPONSES |
|---|---|
| **General comments** | |
| **reconsidered after major revisions:** | Thank you very much for the positive feedback. We continue to strive for the improvement of the manuscript, particularly based on the concerns raised by the reviewer. Please see below our specific responses and refer to the attached revised manuscript for more details. |
| The revised manuscript is significantly improved. The authors were able to address the concerns of the reviewers. | |
| **Major Concerns:** | |
| Thank you for considering my 1st major comment in the literature review part of the paper. | Thank you very much. We have addressed the 1st major comment and we are happy that it has been to your satisfaction. |
| For my second concern, it is now clear that two-way nesting was used, which implies all D1 forcings are different for all sensitivity runs since WRF would not allow having different parameterizations in a two-way nesting run. Kindly address the following: | Yes, that is correct. We have used two-way nesting in the sensitivity runs following recommended practice and previous studies that looked at sensitivities to physics parameterizations in WRF (Wu et al., 2019, Biswas et al., 2014; Li and Pu, 2009; Parker et al., 2017; Spencer and Shaw 2012; Bopape et al., 2021), studies that simulated Typhoon Haiyan in the Philippines (Li et al., 2018; Nakamura et al., 2016) as well as TC cases in other basins (Parker et al., 2018; Mittal et al., 2019; Reddy et al 2020), among others. Please see Table 1 below, containing ample evidence from the literature, to support our decision to employ two-way nesting for the study of tropical cyclones.

 We have tried to address the specific comments as shown below. |
| 1. Include two-way nesting WRF-related sensitivity studies in your literature review. What is the difference between 1-way and 2-way nesting methods? | Thank you for this suggestion. We have added the following discussion in Lines 190 – 199 of the revised manuscript:

 *"Higher-resolution nested model configuration is widely used in numerical weather prediction and regional climate modelling. The main reason for this is because performing high resolution simulation over very large areas (e.g. an entire major oceanic basin) is computationally too expensive (Kueh et al 2019). The communication between the nested domains can be implemented using one-way or two-way nesting. One-way nesting means that the nested domains are run separately and sequentially starting with the outer domain i.e. the model is first run for the outer domain to create and output which is used to supply the inner domain's boundary file. In a two-way nesting configuration, both domains are run simultaneously and interact with each other, so that the highest possible resolution information produced by the innermost domain affects the solutions over the overlapping area of the coarser domains. The input from the coarse outer domain is introduced through the boundary of the fine inner domain, while feedback to the coarse domain occurs all over the inner domain interior, as its values are* |

| | *replaced by combination of fine inner domain values (Alaka et al., 2022; Mure-Ravaud et al 2019; Harris and Duran 2009)."* |
|---|---|
| | We have also added a short discussion on this based on the literature in the revised manuscript (Lines 200 to 215), and also shown below: |
| | *"We have used two-way nesting in the sensitivity runs, rather than one-way nesting, following recommended practice and previous studies that looked at sensitivities to physics parameterizations in WRF (Wu et al., 2019, Biswas et al., 2014; Li and Pu, 2009; Parker et al., 2017; Spencer and Shaw 2012; Bopape et al., 2021), studies that simulated Typhoon Haiyan in the Philippines (Li et al., 2018; Nakamura et al., 2016), as well as TC cases in other basins (Parker et al., 2018; Mittal et al., 2019; Reddy et al 2020), among others. Studies of the differences in using 1-way and 2-way nesting in regional modelling have been, the topic of multiple previous papers (e.g. Spencer and Shaw 2012; Matte et al. 2016; Raffa et al., 2021; Lauwaet et al. 2013; Harris and Durran, 2010, Chen et al 2010; Gao et al., 2019). A comprehensive discussion on the differences and uncertainties associated with 1-way or 2-way nesting can also be found in Harris (2010). Studies such as those of Chen et al (2010) and Gao et al. (2019) have shown that the use of one-way or two-way nesting showed little difference in the results, but some studies showing that two-way nesting improves the simulations of TCs e.g. Typhoon Parma in the Philippines (Spencer and Shaw et al., 2012) and Typhoon Kai-tak (Wu et al., 2019). In addition, previous TC case studies in the Philippines have also used the two-way nesting configuration e.g. Mori et al. (2014), Takayabu et al. (2015), Nakamura et al (2016) and in other TC basins (Parker et al., 2018; Davis et al., 2008; Mittal et al., 2019; Reddy et al., 2020), as well as looking at sensitivity to different physics parameterizations (Wu et al 2019, Biswas et al 2014, Li and Pu 2009) as summarized in Table 1 of the Appendix. Two-way nesting is also used in operational TC forecasting (Mehra et al., 2018) and in the experimental Hurricane WRF system (Zhang et al., 2016) as well as in Convection-Permitting Regional Climate Models (Lucas-Picher et al., 2021)."* |
| 2. Again, even though all runs have the same domain settings (dx,dt,nx,ny etc.), it seems inappropriate and difficult to compare D2 sensitivities on TC track and intensity to model parameterizations if the initial forcing (D1) for KF and TK experiments do not have the same model physics. That is, KF runs have KF D1 forcings, while TK runs have TK D1 forcings. | We want to make it absolutely clear that two-way nesting was used exclusively in our experiments and that 2-way nesting underpins the communication between D1 and D2; we have not used one-way nesting in any of our experiments. One-way forcing still comes, of course, from the outer boundaries of the coarsest domain, where information from the forcing GCM enters the coarsest regional domain, D1. |

The purpose of the coarsest domain, D1, is to mediate the signals originating from the entirely different physics package in the driving GCM (which enters D1 at the outer boundaries), and the physics that we are trying to study in the inner domain. We do not want to introduce a third physical parametrization between GCM and RCM, on the coarse domain, which could in principle be done with 1-way nesting, but is not a modelling strategy commonly adopted in the study of tropical cyclones (see our literature review, and below).

Therefore, the true comparison of physics performance is inside the finest domain, D2, while D1 is acting as more of a mediator. Since we are exclusively using two-way nesting and there is feedback from the outer to the inner domain at each time step and vice versa, it is crucially important that the same physics parameterization is used in both domains, otherwise we would introduce spurious noise at each time step. This is the recommended practice in using WRF with multiple and nested domains (Werner and Wang, 2017; Dudhia 2015) i.e. all physics schemes must be the same for all domains except for when cumulus scheme must be turned off in 3-4km grid intervals (Chen 2022, personal communication, 25 July 2022). Multiple other studies point out the various issues in attempting to employ two-way nesting with differing physics parameterization across the RCM nest boundaries (e.g. in precipitation fields of the mother/outer domain (Dudhia 2015, Warner and Hsu 2000) and used as in past studies e.g. Wang and Wang, 2014; Islam et al., 2015). The use of different cumulus schemes in different domains is also prohibited in WRF.

| How much of the sensitivity is caused by the interactions of D1 and D2 (due to 2-way nesting), and how much from changing model physics/configuration? Kindly show and separate the contribution/effects of the nesting method (1-way vs. 2-way nesting) and parameterizations. | Thank you very much for pointing this out. The uncertainty related to the use of multiple nested grids can result from mismatched model physics across nested-domain boundaries, therefore consistency between nested grids is important (Kueh et al., 2019). A comprehensive discussion on the difference and uncertainties associated with 1-way or 2-way nesting can also be found in Harris (2010), and some limitations in the use of 2-way nesting in regional climate simulations for TCs are discussed in Hashimoto et al (2016). |
| If you perform a 1-way nesting simulation, how much does it differ from its two-way nesting counterpart? May I suggest 1-way nesting experiments as part of the supplementary (initial runs) or major experiments to show the nesting method's influence on the simulations? | As mentioned above, we have chosen to perform a study of TCs with 2-way nesting, because it is the predominant approach in the literature, as well as reflective of our approach in the study of scale interactions, which is only possible with 2-way nesting. Please see Table 1 below with the relevant literature review as well as the added text in Item #1 indicated in this response to support our decision. Studies of the differences in 1-way and 2-way nesting in regional modelling are, instead, the topic of multiple papers already written in the past (e.g. |

| | Spencer and Shaw 2012; Matte et al. 2016; Raffa et al., 2021; Lauwaet et al. 2013; Harris and Durran, 2010, Chen et al 2010; Gao et al., 2019). For example, Spencer and Shaw (2012) showed that the intensity of simulated Typhoon Parma (Philippines) was found to be improved (more accurate) when two-way nesting was used than with one-way nesting. Higher accuracy and efficiency were also shown using two-way nesting in simulating Typhoon Kai-tak (Wu et al., 2019).

We feel that performing 1-way simulations would be substantial additional work and would greatly increase the length of the paper and distract from the main message. We feel that the reviewer is asking for an additional second study, both in terms of the volume of work and the length of the new text+figures, but mostly in terms of the type of study that we would be conducting if we were to re-run all experiments with 1-way nesting. We hope that the discussion above based on literature, and as included in the revised manuscript, would suffice. |
|---|---|
| Since two-away nesting implies D1-D2 interactions, both domain runs should be considered in the sensitivity analysis? What are the results of D1 runs? | Since we have exclusively used two-way nesting, we think that it is not necessary to show the result from the outer domain (D01) since a) in the overlapping region, the results of D01 are overwritten at each time step by the solutions of D02; b) as explained above, D01 is mostly an intermediary between GCM and RCM: as shown in most studies with a similar setup, D01 is used as a means to ensure smooth results for the inner domain (D02).

Nevertheless, for the purpose of this response, we have shown some of results from D01 as indicated in our earlier response to the first round of review (Supplementary Figure 8). But we believe that it is not necessary to show in the final manuscript. |
| **Minor suggestions** | |
| 1. Line 423-425 and other sections with surface roughness length discussion: Kindly check also other references (e.g. Montgomerry et al 2010, Smith et al 2014, their references, and other studies) on the impact of friction (Cd) to tropical cyclones. | Thank you for this suggestion. We have added a short discussion on this based on the literature in the revised manuscript (Lines 291-300), and also shown below:

*"There are limited studies on the sensitivity of TC intensity due to surface heat flux because to a lack of in-situ measurements (Montgomery et al., 2010; Green and Zhang, 2013; Smith et al., 2014), particularly under high-wind conditions (Liu et al., 2014). Emanuel (1986) put forward the idea that TC intensity is proportional to the square root of the ratio of the surface exchange coefficients of enthalpy, and momentum. According to Zhang et al. (2015), increasing surface friction would also increase boundary layer inflow, which would subsequently boost angular momentum convergence and intensify a TC. However, as surface friction also increases the momentum and heat* |

| | |
|---|---|
| | *dissipation to boundary layer winds, this might result in negative impact on TC intensity (Liu et al., 2014). Despite and playing a significant role in surface heat fluxes, Chen et al. (2018) hypothesized that the influence of on TC growth was minimal because it caused moderate sea-surface cooling. Further investigation on these aspects of surface heat flux is required in the future."* |
| 2. For the maps on environmental factors, may I suggest setting the base map to high resolution. If using GrADS, set mpdset hires. | Thank you for this suggestion. We have updated the basemaps to higher resolution and replaced the figures (12,13,14) in the revised manuscript. |
| Thank you and congratulations! | Thank you as well. |

**Table 1 Summary of some studies that used two-way nesting in WRF for tropical cyclone simulations as NWP or RCM**

| Authors and Year | TC Cases | Basin / Region | Resolution (Domains) |
|---|---|---|---|
| Spencer and Shaw 2012* | TY Parma | Western North Pacific (WNP) | 12km (D01); 3k (D02) |
| Li et al 2018* | TY Haiyan | WNP | 18–45 km (D01), 6–15 km (D02), and 2–5 km (D03) |
| Nakamura et al 2016 | TY Haiyan | WNP | not specified |
| Wu et al 2019* | Typhoon Kai-tak | WNP | 15km (D01), 5km (D02) |
| Biswas et al 2014* | Hurricane Katia | Atlantic and Eastern North Pacific | HWRF ~27km (D01), ~9km (D02), ~3 km (D03) |
| Li and Pu 2009* | Hurricane Emily | Atlantic Basin | 27km (D01), 9km (D02), 3 km (D03) |
| Davis et al 2008 | Hurricane Katrina | Atlantic | 12 (D01), 4 (D02) |
| Fierro et al 2009 | Hurricane Rita | Atlantic | 15–5-, 12–4-, 9–3-, 6–2-, and 3–1-km |
| Parker et al 2017* | TC Yasi | Australia | 36 (D01), 12 (D02) |
| Parker et al 2018 | TC Yasi, Ita, Marcia | Australia | 36 (D01), 12 (D02) |
| Bopape et al 2021* | TC Idai | Africa | 6km (D01) |
| Mittal et al 2019 | TC Phailin (2013) | Bay of Bengal | 30 (D01), 10 (D02) |
| Reddy et al 2020 | TC Vardah, Madi, Hudhud and Phailin | Bay of Bengal | 27 (D01), 9 (D02), 3 (D03) |

*Focused on sensitivity studies on different physics parameterizations (including cumulus schemes)*

**References**

Alaka, G. J., Jr., Zhang, X., & Gopalakrishnan, S. G. (2022). High-Definition Hurricanes: Improving Forecasts with Storm-Following Nests, Bulletin of the American Meteorological Society, 103(3), E680-E703. Retrieved Aug 1, 2022, from https://journals.ametsoc.org/view/journals/bams/103/3/BAMS-D-20-0134.1.xml

Bopape, M.-J.M.; Cardoso, H.; Plant, R.S.; Phaduli, E.; Chikoore, H.; Ndarana, T.; Khalau, L.; Rakate, E. Sensitivity of Tropical Cyclone Idai Simulations to Cumulus Parametrization Schemes. Atmosphere 2021, 12, 932. https://doi.org/10.3390/atmos12080932

Biswas, M. K., L. Bernardet, and J. Dudhia (2014), Sensitivity of hurricane forecasts to cumulus parameterizations in the HWRF model, Geophys. Res. Lett., 41, 9113–9119, doi:10.1002/2014GL062071.

Chen, M. (2022). Personal Communication. 25 July 2022.

Chen, S., T. J. Campbell, H. Jin, S. Gaberšek, R. M. Hodur, and P. Martin, 2010: Effect of two-way air–sea coupling in high and low wind speed regimes. Mon. Wea. Rev., 138, 3579–3602. https://apps.dtic.mil/sti/pdfs/ADA530501.pdf

Davis, C., Wang, W., Chen, S. S., Chen, Y., Corbosiero, K., DeMaria, M., ... & Xiao, Q. (2008). Prediction of landfalling hurricanes with the advanced hurricane WRF model. *Monthly weather review*, *136*(6), 1990-2005. Dudhia, J.: Overview of WRF Physics. Retrieved from http://homepages.see.leeds.ac.uk/~lecag/wiser/sample_wiser_files.dir/Physics_Dudhia.ppt.pdf 2015.

Di Z, Gong W, Gan Y, Shen C, Duan Q. Combinatorial Optimization for WRF Physical Parameterization Schemes: A Case Study of Three-Day Typhoon Simulations over the Northwest Pacific Ocean. Atmosphere. 2019; 10(5):233. https://doi.org/10.3390/atmos10050233

Fierro, A. O., Rogers, R. F., Marks, F. D., and Nolan, D. S. (2009). The Impact of Horizontal Grid Spacing on the Microphysical and Kinematic Structures of strong Tropical Cyclones Simulated with the WRF-ARW Model. Monthly Weather Rev. 137, 3717–3743. doi:10.1175/2009mwr2946.1

Gao, K., Harris, L., Chen, J.-H., Lin, S.-J., & Hazelton, A. (2019). Improving AGCM hurricane structure with two-way nesting. Journal of Advances in Modeling Earth Systems, 11, 278– 292. https://doi.org/10.1029/2018MS001359

Harris, L. M. (2010). On the relative performance of one-way and two-way grid nesting. Dissertation, University of Washington.

Harris, L. and Durran, D. (2009). An Idealized Comparison of One-Way and Two-Way Grid Nesting. M onthly Weather Review. American Meteorological Society. Volume 138. https://atmos.uw.edu/~durrand/pdfs/AMS/2010_Harris_Durran_MWR.pdf

Hashimoto, A., Done, J.M., Fowler, L.D. et al. Tropical cyclone activity in nested regional and global grid-refined simulations. Clim Dyn 47, 497–508 (2016). https://doi.org/10.1007/s00382-015-2852-2

Islam, T., Srivastava, P.K., Rico-Ramirez, M.A. et al.: Tracking a tropical cyclone through WRF–ARW simulation and sensitivity of model physics. Nat Hazards 76, 1473–1495, doi:10.1007/s11069-014-1494-8, 2015.

Kueh, M.-T., Chen, W.-M., Sheng, Y.-F., Lin, S. C., Wu, T.-R., Yen, E., Tsai, Y.-L., and Lin, C.-Y.: Effects of horizontal resolution and air–sea flux parameterization on the intensity and structure of simulated Typhoon Haiyan (2013), Nat. Hazards Earth Syst. Sci., 19, 1509–1539, doi:10.5194/nhess-19-1509-2019, 2019.

Lauwaet, D., Viaene, P., Brisson, E., Van Noije, T., Strunk, A., Van Looy, S., … Janssen, S. (2013). Impact of nesting resolution jump on dynamical downscaling ozone concentrations over Belgium. Atmospheric Environment, 67, 46–52. https://doi.org/10.1016/j.atmosenv.2012.10.034

Li X, Pu Z. Sensitivity of Numerical Simulations of the Early Rapid Intensification of Hurricane Emily to Cumulus Parameterization Schemes in Different Model Horizontal Resolutions, Journal of the Meteorological Society of Japan. Ser. II, 2009, Volume 87, Issue 3, Pages 403-421, Released on J-STAGE July 08, 2009, Online ISSN 2186-9057, Print ISSN 0026-1165, https://doi.org/10.2151/jmsj.87.403, https://www.jstage.jst.go.jp/article/jmsj/87/3/87_3_403/_article/-char/en

Li, F., Song, J. & Li, X. A preliminary evaluation of the necessity of using a cumulus parameterization scheme in high-resolution simulations of Typhoon Haiyan (2013). Nat Hazards 92, 647–671 (2018). https://doi.org/10.1007/s11069-018-3218-y

Liu, L., Wang, G., Zhang, Z. et al. Effects of Drag Coefficients on Surface Heat Flux during Typhoon Kalmaegi (2014). Adv. Atmos. Sci. 39, 1501–1518 (2022). https://doi.org/10.1007/s00376-022-1285-1

Lucas-Picher, P., Argüeso, D., Brisson, E., Tramblay, Y., Berg, P., Lemonsu, A., Kotlarski, S., & Caillaud, C. (2021). Convection-permitting modeling with regional climate models: Latest developments and next steps. Wiley Interdisciplinary Reviews: Climate Change, 12( 6), e731. https://doi.org/10.1002/wcc.731

Matte, D., Laprise, R. & Thériault, J.M. Comparison between high-resolution climate simulations using single- and double-nesting approaches within the Big-Brother experimental protocol. *Clim Dyn* **47,** 3613–3626 (2016). https://doi.org/10.1007/s00382-016-3031-9

Mehra, Vijay Tallapragada, Zhan Zhang, Bin Liu, Lin Zhu, Weiguo Wang, Hyun-Sook Kim. (2019). Advancing the State of the Art in Operational Tropical Cyclone Forecasting at Ncep, Tropical Cyclone Research and Review, Volume 7, Issue 1, 2018, Pages 51-56, ISSN 2225-6032, https://doi.org/10.6057/2018TCRR01.06.

Mure-Ravaud M, Kavvas ML, Dib A. Investigation of Intense Precipitation from Tropical Cyclones during the 21st Century by Dynamical Downscaling of CCSM4 RCP 4.5. Int J Environ Res Public Health. 2019 Feb 26;16(5):687. doi: 10.3390/ijerph16050687. PMID: 30813587; PMCID: PMC6427206.

Mittal, R., Tewari, M., Radhakrishnan, C., Ray, P., and Singh, T.: Response of tropical cyclone Phailin (2013) in the Bay of Bengal to climate perturbations. Climate Dynamics, (2013). doi:10.1007/s00382-019-04761,2019.

Mori, N., Kato, M., Kim, S., Mase, H., Shibutani, Y., Takemi, T., Tsuboki, K., and Yasuda, T.: Local amplification of storm surge by Super Typhoon Haiyan in Leyte Gulf. Geophysical Research Letters, Volume 41, Issue 14, Pages 5106-5113. doi:10.1002/2014GL060689, 2014.

Nakamura, R., Shibayama, T., Esteban, M. and Iwamoto, T.: Future typhoon and storm surges under different global warming scenarios: case study of typhoon Haiyan (2013). Nat. Hazards, 82, 1645-1681. doi:10.1007/s11069-016-2259-3, 2016.

Parker CL, Lynch AH, Mooney PA (2017) Factors affecting the trajectory and intensification of tropical cyclone yasi (2011). Atmos Res. https://doi.org/10.1016/j.atmosres.2017.04.002

Parker, C.L., Bruyère, C.L., Mooney, P.A. et al.: The response of land-falling tropical cyclone characteristics to projected climate change in northeast Australia. Clim Dyn 51, 3467–3485, doi:10.1007/s00382-018-4091-9, 2018.

Raffa, M., Reder, A., Adinolfi, M., & Mercogliano, P. (2021). A comparison between one-step and two-step nesting strategy in the dynamical downscaling of regional climate model COSMO-CLM at 2.2 km driven by ERA5 reanalysis. Atmosphere, 12(2), 260. https://doi.org/10.3390/atmos12020260

Reddy, P., Sriram, D., Gunthe, S.S. et alet al.,., (2021). Impact of climate change on intense Bay of Bengal tropical cyclones of the post-monsoom season: a pseudo global warming approach. Clim Dyn 56, 2855–2879. https://doi.org/10.1007/s00382-020-05618-3

Spencer P, Shaw B, Pajuelas B. 2012. Sensitivity of typhoon Parma to various WRF model configurations. In Technical Report on 92nd American Meteorological Society Annual Meeting, 26 January 2012, New Orleans, LA; 608–619. American Meteorological Society: Boston, MA. 2012.

Takayabu, I., Hibino, K., Sasaki, H., Shiogama, H., Mori, N., Shibutani, Y. and Takemi, T.: Climate change effects on the worst-case storm surge: a case study of Typhoon Haiyan, Environ. Res. Lett. 10 089502. doi:10.1088/1748-9326/10/6/064011, 2015.

Warner, T. T. and Hsu, H. M.: Nested-model simulation of moist convection: the impact of coarse-grid parameterized convection on fine-grid resolved convection, Mon. Wea. Rev., 128, 2211–2231, 2000.

Werner, K. and Wang, G: Nesting in WRF. Climate Science Org Australia. Retrieved from /https://www.climatescience.org.au/sites/default/files/werner_nesting.pdf, 2017.

Wu, Z., Jiang, C., Deng, B. et al. Sensitivity of WRF simulated typhoon track and intensity over the South China Sea to horizontal and vertical resolutions. Acta Oceanol. Sin. 38, 74–83 (2019). https://doi.org/10.1007/s13131-019-1459-z

Zhang, X., S. G. Gopalakrishnan, S. Trahan, T. S. Quirino, Q. Liu, Z. Zhang, G. Alaka, and V. Tallapragada, 2016: Representing multiscale interactions in the Hurricane Weather Research and Forecasting modeling system: Design of multiple sets of movable multi-level nesting and the basin-scale HWRF forecast application. Wea. Forecasting, 31, 2019–2034, doi:10.1175/WAF-D-16-0087.1.

**Notification to the authors from the Editorial Team [Validation, 4 July 2022]**

| Notification to the authors | Authors' ACTIONS |
|---|---|
| 1. For the next revision, please rename the supplement`s figures regarding our standards: https://www.natural-hazards-and-earth-system-sciences.net/submission.html#assets / Supplements /. Please note that figures must be numbered sequentially, without reference to the numbering of subsections. | We have renamed the Supplementary Figures (see revised material) and are now numbered sequentially. |
| 2. Regarding of sources of the supplement figures: for the next revision, please check if your figures containing maps/aerial images require a copyright statement/image credit and add it to the figures (or captions) (https://publications.copernicus.org/for_authors/manuscript_preparation.html#mapsaerials). If these figures were entirely created by the authors, there is no need to add a copyright statement or credit. In that case it is important that you confirm this explicitly by email. | All Supplementary Figures were created by the authors. The source of the data in Supplementary Figure 9a has been provided in the figure title. |
| 3. Regarding the figure #8: for next revision, please add the sources from footnote to the figure`s caption. | Revised in Lines 531-532 on the revised manuscript |

---

## Author Response (AR3)

*Response to comments of **Anonymous Referee 2** on*

**Sensitivity of simulating Typhoon Haiyan (2013) using WRF: the role of cumulus convection, surface flux parameterizations, spectral nudging, and initial and boundary conditions**

Delfino et. al.

**Anonymous Referee #2 [Report #1 Submitted on 29 Aug 2022]**

| Editor's COMMENTS | Authors' RESPONSES |
|---|---|
| Dear Authors,

I am glad to inform you that your paper can be now accepted subject to the technical corrections proposed by the reviewer. Please correct the manuscript accordingly. Thank you for considering NHESS for the publication of your research. | Dear Editor,

Thank you very much for giving us the opportunity to publish our research at NHESS. We have made the suggested technical corrections of Reviewer 2 in the revised manuscript. All line numbers refer to the revised manuscript. |
| **Reviewer 2's COMMENTS** | **Authors' RESPONSES** |
| **General comments** | |
| **accepted subject to technical corrections.** | Thank you very much for the positive review. We have made the suggested technical corrections in the revised manuscript. Please see below our specific responses and refer to the revised manuscript for more details. |
| Thank you for considering all the reviewers' concerns. The revised manuscript has improved a lot and should be accepted for publication with minimal revisions. | |
| I think Figure 8 in the supplementary materials somehow shows the negligible impact of two-way nesting in comparison to model parameterizations in your case study. The reviewer was concerned with the distinction between the impact of model parameterizations and domain interactions due to two-way nesting.

Kindly refer to the supplementary figures in the main text rather than saying "not shown here". | Thank you for pointing this out. We have referred to the Supplementary Figures in the main text, particularly: Supplementary Figures 1-5 on domain configuration (Line 218); Supplementary Figures 6-7 on different initial times (Line 225); and Supplementary Figure 8 on two-way nesting and analysis of the inner domain only (Line 236). |
| **Minor concerns:** | |
| Line 58 – Villafuerte uses a regional climate model. | We were referring to the need for using RCMs/LAMs as emphasized by Villafuerte at al 2021, but nevertheless, we have removed this reference in this line (58) to avoid confusion. |

| | |
|---|---|
| Line 294: Zhang et al (2015) is not in the list of references. | Thank you for spotting this. We have added it in the references of the revised manuscript. |
| Line 362-365: Kindly clarify. TK is deviated by 50 km at 36h, while KF deviated by 50 km during the first 36 hrs? Please clarify if both schemes deviated by 50km in the first 36 hrs? | TK simulations deviation were LESS THAN 50km, ave=18km (indicated in Line 367 of the revised manuscript) while KF simulations were MORE THAN 50 km, ave=61.5km (indicated in Line 369 of the revised manuscript). |
| Line 373: Write Figure 3 to Figure 3a. May I suggest putting Line 377 first before Line 373? | Thank you for this suggestion. The text reads better that way. We have moved Line 384 to 378 in the revised manuscript. |
| Line 390. TK is less dependent on nudging … then Line 392 seems to contradict it. | Thank you for the clarification. We have revised this in Line 395 of the revised manuscript. |
| Line 393: Statistically significant? | Not statistically significant and we have indicated the value in Line 397 of the revised manuscript. |
| Line 404-405. Simplify the sentence. ", this is a difference" looks like another sentence. | Thank you for spotting this complicated sentence. We have revised this in Lines 409-411 in the revised manuscript. |
| Line 407: ", however, this". What does "this" refers to? Kindly rewrite. | Refers to the simulations using KF scheme and we have revised in the manuscript (Lines 412-413) . |
| Line 422: Statistically significant? By how much? | Yes, this is statistically significant and we have indicated the value in line 428 in the revised manuscript. |
| Lines 434-435: Simplify. The two sentences have basically the same meaning. | Thank you for spotting this. We have removed the second sentence and retained the first (Lines 440-441) in the revised manuscript. |
| Line 462: How about sf2? | Indicated in Lines 468-469 in the revised manuscript. |
| Line 650 : "at 25 km and 5 km". Do you mean 5 km only since the analysis was done only with the finest domain? | The analysis was done only for the 5km inner domain. We have revised this in Line 656 of the revised manuscript |
| Kindly improve and simplify sentence constructions. Avoid repeating a word or a thought several times in one sentence:

 e.g. Line 646-649:

 E.g. "As climate models project more intense storms, such as Typhoon Haiyan, will occur more frequently in the future due to climate change, it is important to improve their representation in high-resolution models, in order to improve understanding of TCs under climate change and improve confidence in model projections, and more importantly, for risk and impact assessments." | Apologies for these. We have improved these in Lines 652-655 and Lines 675-678 in the revised manuscript. |
| Congratulations to the authors for doing a great job and sharing their research with the community. | Thank you very much for your valuable inputs and for letting us share our research. |